# Decentralized $Q$-Learning in Zero-sum Markov Games

**Muhammed O. Sayin**[*]
Bilkent University
sayin@ee.bilkent.edu.tr

**Kaiqing Zhang**[*]
MIT
kaiqing@mit.edu

**David S. Leslie**
Lancaster University
d.leslie@lancaster.ac.uk

**Tamer Başar**
UIUC
basar1@illinois.edu

**Asuman Ozdaglar**
MIT
asuman@mit.edu

## Abstract

We study multi-agent reinforcement learning (MARL) in infinite-horizon discounted zero-sum Markov games. We focus on the practical but challenging setting of *decentralized* MARL, where agents make decisions without coordination by a centralized controller, but only based on their own payoffs and local actions executed. The agents need not observe the opponent's actions or payoffs, possibly being even oblivious to the presence of the opponent, nor be aware of the zero-sum structure of the underlying game, a setting also referred to as *radically uncoupled* in the literature of learning in games. In this paper, we develop a radically uncoupled $Q$-learning dynamics that is both *rational* and *convergent*: the learning dynamics converges to the best response to the opponent's strategy when the opponent follows an asymptotically stationary strategy; when both agents adopt the learning dynamics, they converge to the Nash equilibrium of the game. The key challenge in this decentralized setting is the *non-stationarity* of the environment from an agent's perspective, since both her own payoffs and the system evolution depend on the actions of other agents, and each agent adapts her policies simultaneously and independently. To address this issue, we develop a two-timescale learning dynamics where each agent updates her local $Q$-function and value function estimates concurrently, with the latter happening at a slower timescale.

## 1 Introduction

Reinforcement learning (RL) has achieved tremendous successes recently in a wide range of applications, including playing the game of Go [Silver et al., 2017], playing video games (e.g., Atari [Mnih et al., 2015] and Starcraft [Vinyals et al., 2019]), robotics [Lillicrap et al., 2015, Kober et al., 2013], and autonomous driving [Shalev-Shwartz et al., 2016, Sallab et al., 2017]. Most of these applications involve *multiple* decision-makers, where the agents' rewards and the evolution of the system are affected by the joint behaviors of all agents. This setting naturally leads to the problem of multi-agent RL (MARL). In fact, MARL is arguably one key ingredient of large-scale and reliable autonomy, and is significantly more challenging to analyze than single-agent RL. There has been a surging interest recently in both a deeper theoretical and empirical understanding of MARL; see comprehensive overviews on this topic in Busoniu et al. [2008], Zhang et al. [2021], Hernandez-Leal et al. [2019].

The pioneering work that initiated the sub-area of MARL, where the model of Markov/stochastic games [Shapley, 1953] has been considered as a framework, is Littman [1994]. Since then, there

---

[*]Equal contribution.

35th Conference on Neural Information Processing Systems (NeurIPS 2021).

has been a plethora of works on MARL in Markov games; see a detailed literature review in the supplementary material. These algorithms can be broadly categorized into two types: *centralized/coordinated* and *decentralized/independent* ones. For the former type, it is assumed that there exists a central controller for the agents, who can access the agents joint actions and local observations. With full awareness of the game setup, the central controller coordinates the agents to optimize their own policies, and aims to compute an equilibrium. This centralized paradigm is typically suitable for the scenarios when a simulator of the game is accessible [Silver et al., 2017, Vinyals et al., 2017]. Most existing MARL algorithms in Markov games have focused on this paradigm.

Nevertheless, in many practical multi-agent learning scenarios, e.g., multi-robot control [Wang and de Silva, 2008], urban traffic control [Kuyer et al., 2008], as well as economics with rational decision-makers [Fudenberg and Levine, 1998], agents make decentralized decisions *without* a coordinator. Specifically, agents make updates independently with only *local* observations of their own payoff and action histories, usually in a myopic fashion. Besides its ubiquity in practice, this decentralized paradigm also has the advantage of being *scalable*, as each agent only cares about her own policy and/or value functions, and the algorithms complexity do not suffer from the exponential dependence on the number of agents.

Unfortunately, establishing provably convergent decentralized MARL algorithms is well-known to be challenging; see the non-convergent cases (even in the fully cooperative setting) in Tan [1993], Boutilier [1996], Claus and Boutilier [1998], and see Matignon et al. [2012] for more empirical evidences. The key challenge is the *non-stationarity* of the environment from each agent's perspective since all agents are adapting their policies simultaneously and independently. In other words, the opponent is not playing according to a stationary strategy. This non-stationarity issue is in fact one of the core issues in (decentralized) MARL [Busoniu et al., 2008, Hernandez-Leal et al., 2017].

Studying when self-interested players can converge to an equilibrium through non-equilibrium adaptation is core question in the related literature of learning in games [Fudenberg and Levine, 1998, 2009]. For example, simple and stylized learning dynamics, such as fictitious play, are shown to converge to an equilibrium in certain but important classes of games, e.g., zero-sum [Robinson, 1951, Harris, 1998] and common-interest [Monderer and Shapley, 1996], in repeated play of the same game. However, we cannot generalize these results to decentralized MARL in *Markov* games (also known as stochastic games, introduced by Shapley [1953]), because agents strategies affect not only the immediate reward, as in the repeated play of the same strategic-form game, but also the rewards that will be received in the future. Therefore, the configuration of the induced *stage games* are not necessarily stationary in Markov games.

**Contributions.** In this paper, we present a provably convergent decentralized MARL learning dynamics[2] for zero-sum discounted Markov games over an infinite horizon with minimal information available to agents. Particularly, each agent only has access to her immediate reward and the current state with perfect recall. They do not have access to the immediate reward the opponent receives. They do not know a model of their reward functions and the underlying state transitions probabilities. They are oblivious to the zero-sum structure of the underlying game. They also do not observe the opponent's actions. Indeed, they may even be oblivious to the presence of other agents. Learning dynamics with such minimal information is also referred to as being *radically uncoupled* or *value-based* in the literature of learning in games [Foster and Young, 2006, Leslie and Collins, 2005].

To address the non-stationarity issue, we advocate a two-timescale adaptation of the individual $Q$-learning, introduced by Leslie and Collins [2005] and originating in Fudenberg and Levine [1998]. Particularly, each agent infers the opponent's strategy indirectly through an estimate of the local $Q$-function (a function of the opponent's strategy) and simultaneously forms an estimate of the value function to infer the continuation payoff. The slow update of the value function estimate is natural since agents tend to change their strategies faster than their estimates (as observed in the evolutionary game theory literature, e.g., Ely and Yilankaya [2001], Sandholm [2001]), but this also helps weakening the dependence between the configuration of the stage games (specifically the global $Q$-functions) and the strategies. We show the almost sure convergence of the learning dynamics to the Nash equilibrium using the stochastic approximation theory, by developing a novel Lyapunov function and identifying the sufficient conditions precisely later in §3. Our techniques

---

[2]To emphasize the difference from many existing MARL algorithms that focus on the *computation* of Nash equilibrium, we refer to our update rule as *learning dynamics*, following the literature of learning in games.

toward addressing these challenges might be of independent interest. We also verify the convergence of the learning dynamics via numerical examples.

To the best of our knowledge, our learning dynamics appears to be one of the first provably convergent decentralized MARL learning dynamics for Markov games that enjoy all the appealing properties below, addressing an important open question in the literature [Pérolat et al., 2018, Daskalakis et al., 2020]. In particular, our learning dynamics –

- requires only minimal information available to the agents, i.e., it is a radically uncoupled learning dynamic, unlike many other MARL algorithm, e.g., Pérolat et al. [2015], Sidford et al. [2019], Leslie et al. [2020], Sayin et al. [2020], Bai and Jin [2020], Xie et al. [2020], Zhang et al. [2020], Liu et al. [2020], Shah et al. [2020];
- requires no coordination or communication between agents during learning. For example, agents always play the (smoothed) best response consistent with their self-interested decision-making, contrary to being coordinated to keep playing the same strategy within certain time intervals as in Arslan and Yuksel [2017] and Wei et al. [2021];
- requires no *asymmetric* update rules and/or stepsizes for the agents unlike existing literature [Vrieze and Tijs, 1982], [Bowling and Veloso, 2002], [Leslie and Collins, 2003], [Daskalakis et al., 2020], [Zhao et al., 2021, Guo et al., 2021]. Such an asymmetry implies implicit coordination between agents to decide who follows which update rule or who chooses which stepsize (and correspondingly who reacts fast or slow). Daskalakis et al. [2020] refers to each agent playing a symmetric role in learning as *strongly independent* learning.
- is both *rational* and *convergent*, a desired property for MARL (independent of whether it is centralized or decentralized), e.g., see Bowling and Veloso [2001], Busoniu et al. [2008]. A MARL algorithm is rational if each agent can converge to best-response, when the opponent plays an (asymptotically) stationary strategy; and it can converge only to an equilibrium when all agents adopt it.

A detailed literature review is deferred to the supplementary material due to space limitations. Of particular relevance are two recent works Tian et al. [2020] and Wei et al. [2021] studied decentralized setting similar to ours. Tian et al. [2020] focused on the exploration aspect for finite-horizon settings, and focused on minimizing a weak notion of regret without providing convergence guarantees under self-play.[3] Wei et al. [2021] presented an optimistic variant of the gradient descent-ascent method that shares similar desired properties with our learning dynamics, with a strong guarantee of last-iterate convergence rates. However, the algorithm is delicately designed and different from the common value/policy-based RL update rules, e.g., $Q$-learning, as in our work. Moreover, to characterize finite-time convergence, in the model-free setting, the agents need to coordinate to interact multiple steps at each iteration of the algorithm, while our learning dynamics is coordination-free with natural update rules. These two works can thus be viewed as orthogonal to ours. After submitting our paper, we became aware of a concurrent and independent work Guo et al. [2021], which also developed a decentralized algorithm for zero-sum Markov games with function approximation and finite-sample guarantees. In contrast to our learning dynamics, the algorithm requires a double-loop update rule, and thus is asymmetric and requires coordination between agents. The assumptions and technical novelties in both works are also fundamentally different. See §A.2 for a detailed comparison.

**Organization.** The rest of the paper is organized as follows. We describe Markov games and our decentralized $Q$-learning dynamics in §2. In §3, we present the assumptions and the convergence results. In §4, we provide numerical examples. We conclude the paper with some remarks in §5. The supplementary material includes a detailed literature review and the proofs of technical results.

**Notations.** Superscript denotes player identity. We represent the entries of vectors $q$ (or matrices $Q$) via $q[i]$ (or $Q[i, j]$). For two vectors $x$ and $y$, the inner product is denoted by $x \cdot y = x^T y$. For a finite set $A$, we denote the probability simplex over $A$ by $\Delta(A)$.

---

[3]Note that the same update rule with different stepsize and bonus choices and a certified policy technique, however, can return a non-Markovian approximate Nash equilibrium policy pair in the self-play setting; see Bai et al. [2020] for more details.

## 2 Decentralized $Q$-learning in Zero-sum Markov Games

This section presents a decentralized $Q$-learning dynamics that does not need access to the opponent's actions and *does not* need to know the zero-sum structure of the underlying Markov game. To this end, we first start by providing a formal description of Markov games.

Consider two players interacting with each other in a common dynamic environment, with totally conflicting objectives. The setting can be described by a two-player zero-sum Markov game, characterized by a tuple $\langle S, \{A_s^i\}_{(i,s)\in\{1,2\}\times S}, \{r_s^i\}_{(i,s)\in\{1,2\}\times S}, p, \gamma\rangle$, where $S$ denotes the set of states, $A_s^i$ denotes the action set of player $i$ at state $s \in S$, and $\gamma \in [0,1)$ denotes the discount factor. At each interaction round, player $i$ receives a reward according to the function $r_s^i : A_s^1 \times A_s^2 \to \mathbb{R}$. Since it is a zero-sum game, we have $r_s^1(a^1, a^2) + r_s^2(a^1, a^2) = 0$ for each joint action pair $(a^1, a^2) \in A_s^1 \times A_s^2$. We denote the transition probability from state $s$ to state $s'$ given a joint action profile $(a_s^1, a_s^2)$ by $p(s'|s, a_s^1, a_s^2)$. Let us denote the stationary (Markov) strategy of player $i$ by $\pi^i := \{\pi_s^i \in \Delta(A_s^i)\}_{s\in S}$. We define the expected utility of player $i$ under the strategy profile $(\pi^1, \pi^2)$ as the expected discounted sum of the reward he collects over an infinite horizon

$$U^i(\pi^1, \pi^2) = \mathbb{E}_{\{a_k^j \sim \pi_{s_k}^j\}_{j\in\{1,2\}}} \left\{ \sum_{k=0}^{\infty} \gamma^k r_{s_k}^i(a_k^1, a_k^2) \right\}, \tag{1}$$

where $\{s_0 \sim p_o, s_{k+1} \sim p(\cdot|s_k, a_k^1, a_k^2), k \geq 0\}$ is a stochastic process describing the evolution of the state over time and $p_o \in \Delta(S)$ is the initial state distribution. The expectation is taken with respect to the initial state, randomness induced by state transitions and mixed strategies.

A strategy profile $(\pi_*^1, \pi_*^2)$ is an $\varepsilon$-Nash equilibrium of the Markov game with $\varepsilon \geq 0$ provided that

$$U^1(\pi_*^1, \pi_*^2) \geq U^1(\pi^1, \pi_*^2) - \varepsilon, \ \forall \pi^1 \tag{2a}$$

$$U^2(\pi_*^1, \pi_*^2) \geq U^2(\pi_*^1, \pi^2) - \varepsilon, \ \forall \pi^2. \tag{2b}$$

A Nash equilibrium is an $\varepsilon$-Nash equilibrium with $\varepsilon = 0$. It is known that such a Nash equilibrium exists for discounted Markov games [Fink, 1964, Filar and Vrieze, 2012].

Given a strategy profile $\pi := (\pi^1, \pi^2)$, we define the value function of player $i$ by

$$v_\pi^i(s) = \mathbb{E}_{\{a_k^j \sim \pi_{s_k}^j\}_{j\in\{1,2\}}} \left\{ r_s^i(a^1, a^2) + \gamma \sum_{s'\in S} v_\pi^i(s')p(s'|s, a^1, a^2) \right\}, \quad \forall s. \tag{3}$$

Note that $U^i(\pi^1, \pi^2) = \mathbb{E}\left\{v_\pi^i(s_0) \mid s_0 \sim p_o\right\}$. We also define the $Q$-function that represents the value obtained for a given state and joint action pair as

$$Q_\pi^i(s, a^1, a^2) = r_s^i(a^1, a^2) + \gamma \sum_{s'\in S} v_\pi^i(s')p(s'|s, a^1, a^2), \quad \forall(s, a^1, a^2), \tag{4}$$

as well as the *local $Q$-function* for player $i$ as

$$q_\pi^i(s, a^i) := \mathbb{E}_{a^{-i} \sim \pi_s^{-i}}\left\{Q_\pi^i(s, a^1, a^2)\right\}, \quad \forall(s, a^i), \tag{5}$$

where $-i$ denotes the opponent of player $i$.

By the one-stage deviation principle, we can interpret the interaction between the players at each stage as they are playing an auxiliary stage game, in which the payoff functions are equal to the $Q$-functions, e.g., see Shapley [1953]. However, the $Q$-functions, and correspondingly the payoff functions in these auxiliary stage games, change with evolving strategies of the players. Therefore, the plethora of existing results for repeated play of the same strategic-form game (e.g., see the review Fudenberg and Levine [2009]) do not generalize here. To address this challenge, we next introduce our decentralized $Q$-learning dynamics.

### 2.1 Decentralized $Q$-learning Dynamics

In our decentralized $Q$-learning dynamics, minimal information is available to players. In other words, they only have access to the immediate reward received and current state visited with perfect recall. They do not observe the actions taken by the opponent. Correspondingly, they cannot form

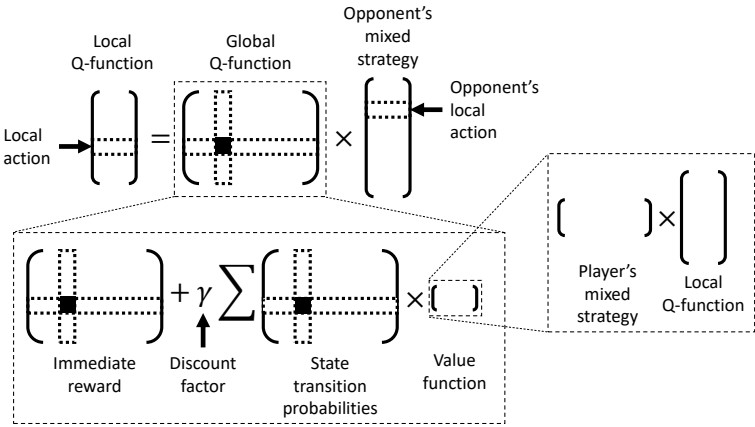

Figure 1: A figurative illustration of the dependence between local $Q$-function, global $Q$-function, opponent's mixed strategy and the value function. From an agents' perspective, the local $Q$-function gives the expected values associated with local actions. Here, agents will infer the opponent's strategy over the local $Q$-function and estimate the value function at a slower timescale to ensure that the *implicit* global $Q$-function is relatively *stationary* compared to the opponent's strategy.

a belief about the opponent's strategy based on the empirical play as in fictitious play [Fudenberg and Levine, 1998] or its variant for stochastic games [Sayin et al., 2020]. Instead, the players can look for inferring the opponent's strategy, e.g., by estimating the local $Q$-function since the local $Q$-function contains information about the opponent's strategy, as illustrated in Figure 1. As seen in Figure 1, however, the local $Q$-function also depends on the global $Q$-function while the global $Q$-function is not necessarily stationary since it depends on the value function, and therefore, depends on the players' evolving strategies. An estimate of the global $Q$ function which is slowly evolving would make it relatively stationary compared to the strategies. However, the players cannot estimate the global $Q$-function directly since they do not have access to the opponent's actions. Instead, they estimate the value function while updating it at a slower timescale. The slow update of the value function estimate makes the implicit global $Q$-function relatively stationary compared to the strategies. Therefore, the players can use the local $Q$-function estimate to infer the opponent's strategy.

Note that the local $Q$-function estimate for different actions would get updated asynchronously via the classical $Q$-learning algorithm since they would be updated only when the associated action is taken, however, the actions are likely to be taken at different frequencies. Instead, here the players update the local $Q$-function estimate via a learning dynamics inspired from the individual $Q$-learning. The individual $Q$-learning, presented by Leslie and Collins [2005] and originating in Fudenberg and Levine [1998], is based on the $Q$-learning with soft-max exploration while the step sizes are normalized with the probability of the actions taken. This normalization ensures that the estimates for every action get updated at the same learning rate in the expectation. We elaborate further on this after we introduce the precise update of the local $Q$-function estimate later in this section.

Each player $i$ keeps track of $\{\hat{q}^i_{s,k}\}_{s \in S}$ and $\{\hat{v}^i_{s,k}\}_{s \in S}$ estimating, respectively, the local $Q$-function and the value function. Player $i$ updates $\{\hat{q}^i_{s,k}\}_{s \in S}$ at a faster timescale than $\{\hat{v}^i_{s,k}\}_{s \in S}$. Players also count the number of times each state $s$ is visited (until the current stage), denoted by $\#s$.

We assume players know that the reward function takes values in $[-R, R]$ for some $R \in (0, \infty)$, i.e., $|r^i_s(a^1, a^2)| \leq R$ for all $(i, s, a^1, a^2)$. Therefore, player $i$ knows that his local $Q$-function $\|q^i_\pi(s, \cdot)\|_\infty \leq R/(1 - \gamma) =: D$ and the value function $|v^i_\pi(s)| \leq D$ for any strategy profile $\pi$ and $s \in S$. Correspondingly, the players initiate these estimates arbitrarily such that $\|\hat{q}^i_{s,0}\|_\infty \leq D$ and $|\hat{v}^i_{s,0}| \leq D$, for all $s$.

Let $s_k$ denote the current state at stage $k$. Player $i$ takes his action $a^i_k \in A^i_{s_k}$, drawn from the smoothed best response $\overline{\mathrm{Br}}^i_{s_k}(\hat{q}^i_{s_k,k}, \tau_{\#s_k}) \in \Delta(A^i_s)$, which depends on a temperature parameter $\tau_{\#s} > 0$ associated with $\#s$. We define

$$\overline{\mathrm{Br}}^i_s(q, \tau) := \operatorname*{argmax}_{\mu \in \Delta(A^i_s)} \left\{ \mu \cdot q + \tau \nu^i_s(\mu) \right\}, \tag{6}$$

Table 1: Decentralized $Q$-learning dynamics in Markov games

---

**Require:** Keep track of $\{\hat{q}^i_{s,k}, \hat{v}^i_{s,k}, \#s\}_{s\in S}$.

1: **Observe the current state** $s_k$.
2: **Update** the entry of **the local $Q$-function estimate** for the previous state $s_{k-1}$ and local action $a^i_{k-1}$ according to

$$\hat{q}^i_{s_{k-1},k}[a^i_{k-1}] = \hat{q}^i_{s_{k-1},k-1}[a^i_{k-1}] + \bar{\alpha}^i_{k-1}\left(r^i_{k-1} + \gamma\hat{v}^i_{s_k,k-1} - \hat{q}^i_{s_{k-1},k-1}[a^i_{k-1}]\right),$$

where $\bar{\alpha}^i_{k-1} = \min\left\{1, \frac{\alpha_{\#s_{k-1}}}{\bar{\pi}^i_{k-1}[a^i_{k-1}]}\right\}$, and $\hat{q}^i_{s,k}[a^i] = \hat{q}^i_{s,k-1}[a^i]$ for all $(s, a^i) \neq (s_{k-1}, a^i_{k-1})$.
3: **Increment** the counter only for the current state by one.
4: **Take action** $a^i_k \sim \bar{\pi}^i_k$ where

$$\bar{\pi}^i_k = \operatorname*{argmax}_{\mu\in\Delta(A^i_{s_k})}\left\{\mu\cdot\hat{q}^i_{s_k,k} + \tau_{\#s_k}\nu^i_{s_k}(\mu)\right\}.$$

5: **Collect the local reward** $r^i_k$ and **update the value function estimate** for the current state according to

$$\hat{v}^i_{s_k,k+1} = \hat{v}^i_{s_k,k} + \beta_{\#s_k}\left[\bar{\pi}^i_k\cdot\hat{q}^i_{s_k,k} - \hat{v}^i_{s_k,k}\right].$$

On the other hand, $\hat{v}^i_{s,k+1} = \hat{v}^i_{s,k}$ for all $s \neq s_k$.

---

where $\nu^i_s$ is a smooth and strictly concave function whose gradient is unbounded at the boundary of the simplex $\Delta(A^i_s)$ [Fudenberg and Levine, 1998]. The temperature parameter $\tau > 0$ controls the amount of perturbation on the smoothed best response. The smooth perturbation ensures that there exists a unique maximizer $\bar{\pi}^i_k := \overline{\mathrm{Br}}^i_{s_k}(\hat{q}^i_{s_k,k}, \tau_{\#s_k})$. Choosing $\nu^i_s(\mu) := -\sum_{a^i\in A^i_s}\mu[a^i]\log\mu[a^i]$ results in the explicit characterization:

$$\bar{\pi}^i_k[a^i] = \frac{\exp\left(\hat{q}^i_{s_k,k}[a^i]/\tau_{\#s_k}\right)}{\sum_{\tilde{a}^i}\exp\left(\hat{q}^i_{s_k,k}[\tilde{a}^i]/\tau_{\#s_k}\right)} > 0. \tag{7}$$

We let player $i$ update his local $Q$-function estimate's entry associated with the current state and local action pair $(s_k, a^i_k)$ towards the reward received plus the discounted continuation payoff estimate. To this end, we include $\hat{v}^i_{s_{k+1},k}$ as an unbiased estimate of the continuation payoff obtained by looking one stage ahead, as in the classical $Q$-learning introduced by Watkins and Dayan [1992]. Due to the one-stage look ahead, the update for the current state and local action can take place just after the game visits the next state. The update of $\hat{q}^i_{s_k,k}$ is given by

$$\hat{q}^i_{s_k,k+1}[a^i_k] = \hat{q}^i_{s_k,k}[a^i_k] + \bar{\alpha}^i_k\left(r^i_k + \gamma\hat{v}^i_{s_{k+1},k} - \hat{q}^i_{s_k,k}[a^i_k]\right), \tag{8}$$

where $\bar{\alpha}^i_k \in (0, 1]$ is defined by $\bar{\alpha}^i_k := \min\left\{1, \frac{\alpha_{\#s_k}}{\bar{\pi}^i_k[a^i_k]}\right\}$, with $\{\alpha_c\}_{c>0}$ a step size sequence, and $r^i_k \in [-R, R]$ denotes the immediate reward of player $i$ at stage $k$. There is no update on others, i.e., $\hat{q}^i_{s,k+1}[a^i] = \hat{q}^i_{s,k}[a^i]$ for all $(s, a^i) \neq (s_k, a^i_k)$. Inspired by the approach in Leslie and Collins [2005], the normalization addresses the asynchronous update of the entries of the local $Q$-function estimate and ensures that every entry of the local $Q$-function estimate is updated at the same rate in expectation. We will show this explicitly in the proof of main theorem in the supplementary material.

Simultaneous to updating the local $Q$-function estimate, player $i$ updates his value function estimate $\hat{v}^i_{s_k,k}$ towards $\bar{\pi}^i_k\cdot\hat{q}^i_{s_k,k}$ corresponding to the expected value of the current state. However, the player uses a different step size $\{\beta_c\}_{c>0}$ and updates $\hat{v}^i_{s_k,k}$ according to

$$\hat{v}^i_{s_k,k+1} = \hat{v}^i_{s_k,k} + \beta_{\#s_k}\left[\bar{\pi}^i_k\cdot\hat{q}^i_{s_k,k} - \hat{v}^i_{s_k,k}\right]. \tag{9}$$

For other states $s \neq s_k$, there is no update on the value function estimate, i.e., $\hat{v}^i_{s,k+1} = \hat{v}^i_{s,k}$. To sum up, player $i$ follows the learning dynamics in Table 1. We emphasize that this dynamic is radically uncoupled since each player's update rule does not depend on the opponent's payoffs or actions. In the next section, we study its convergence properties.

# 3 Convergence Results

We study whether the value function estimates in the learning dynamics, described in Table 1, converge to an equilibrium value of the zero-sum Markov game. The answer is affirmative under certain conditions provided below precisely. The first assumption (with two parts) is related to the step sizes and the temperature parameter, and not to the properties of the Markov game model.

**Assumption 1-i.** *The sequences $\{\alpha_c \in (0,1)\}_{c>0}$ and $\{\beta_c \in (0,1)\}_{c>0}$ are non-increasing and satisfy $\sum_{c=1}^{\infty} \alpha_c = \infty$, $\sum_{c=1}^{\infty} \beta_c = \infty$, and $\lim_{c\to\infty} \alpha_c = \lim_{c\to\infty} \beta_c = 0$.*

**Assumption 1-ii.** *Given any $M \in (0,1)$, there exists a non-decreasing polynomial function $C(\cdot)$ (which may depend on $M$) such that for any $\lambda \in (0,1)$ if $\left\{\ell \in \mathbb{Z}_+ \,|\, \ell \leq c \text{ and } \frac{\beta_\ell}{\alpha_c} > \lambda\right\} \neq \varnothing$, then*

$$\max\left\{\ell \in \mathbb{Z}_+ \,|\, \ell \leq c \quad and \quad \frac{\beta_\ell}{\alpha_c} > \lambda\right\} \leq Mc, \quad \forall c \geq C\left(\lambda^{-1}\right). \tag{10}$$

Assumption 1-i is a common assumption used in stochastic approximation theory, e.g., see Benaim [1999], Borkar [2008]. On the other hand, Assumption 1-ii imposes further condition on the step sizes than the usual two-timescale learning assumption, e.g., $\lim_{c\to\infty} \frac{\beta_c}{\alpha_c} = 0$, to address the asynchronous update of the iterates. Particularly, the iterates evolving at fast timescale can lag behind even the iterates evolving at slow timescale due to their asynchronous update. Assumption 1-ii ensures that this can be tolerated when states are visited at comparable frequencies.

For example, $\alpha_c = c^{-\rho_\alpha}$ and $\beta_c = c^{-\rho_\beta}$, where $0.5 < \rho_\alpha < \rho_\beta \leq 1$, satisfy Assumption 1 since it can be shown that there exists a non-decreasing polynomial, e.g., $C(x) = M_o x^m$, for all $x \geq 1$, where $M_o := M^{-\rho_\beta/(\rho_\beta - \rho_\alpha)} > 0$ and $m \in \{m' \in \mathbb{Z}_+ : m' \geq 1/(\rho_\beta - \rho_\alpha)\}$. We provide the relevant technical details in the supplementary material.

Such learning dynamics is not guaranteed to converge to an equilibrium in every class of zero-sum Markov games. For example, the underlying Markov chain may have an absorbing state such that once the game reaches that state, it stays there forever. Then, the players will not have a chance to improve their estimates for other states. Therefore, in the following, we identify two sets of assumptions (in addition to Assumption 1) imposing increasingly stronger conditions on the underlying game while resulting in different convergence guarantees.

**Assumption 2-i.** *Given any pair of states $(s, s')$, there exists at least one sequence of actions such that $s'$ is reachable from $s$ with some positive probability within a finite number, $n$, of stages.*

**Assumption 2-ii.** *The sequence $\{\tau_c\}_{c>0}$ is non-increasing and satisfies $\lim_{c\to\infty}(\tau_{c+1} - \tau_c)/\alpha_c = 0$ and $\lim_{c\to\infty} \tau_c = \epsilon$ for some $\epsilon > 0$. The step size $\{\alpha_c\}_{c>0}$ satisfies $\sum_{c=1}^{\infty} \alpha_c^2 < \infty$.*

In Assumption 2, we do not let the temperature parameter go to zero. Next we let $\lim_{c\to\infty} \tau_c = 0$ but make the following assumption, imposing further condition on the underlying game and $\{\alpha_c\}_{c>0}$ compared to Assumption 2 to ensure that each state gets visited infinitely often at comparable frequencies and the normalization in the update of the local $Q$-function estimate does not cause an issue since it can be arbitrarily small when $\tau_c \to 0$.

**Assumption 2'-i.** *Given any pair of states $(s, s')$ and any infinite sequence of actions, $s'$ is reachable from $s$ with some positive probability within a finite number, $n$, of stages.*

**Assumption 2'-ii.** *The sequence $\{\tau_c\}_{c>0}$ is non-increasing and satisfies $\lim_{c\to\infty}(\tau_{c+1} - \tau_c)/\alpha_c = 0$ and $\lim_{c\to\infty} \tau_c = 0$. The step size $\{\alpha_c\}_{c>0}$ satisfies $\sum_{c=1}^{\infty} \alpha_c^{2-\rho} < \infty$, for some $\rho \in (0,1)$. There exists $C, C' \in (0, \infty)$ such that $\alpha_c^\rho \exp\left(4D/\tau_c\right) \leq C'$ for all $c \geq C$.*

While being stronger than Assumption 2-i, Assumption 2'-i is still weaker than those used in Leslie et al. [2020]. In Leslie et al. [2020], it is assumed that there is a positive probability of reaching from any state to any other state in *one* stage for any joint action taken by the players. On the other hand, we say that a Markov game is *irreducible* if given any *pure* stationary strategy profile, the states visited form an irreducible Markov chain [Hoffman and Karp, 1966, Brafman and Tennenholtz, 2002]. Assumption 2'-i is akin to the *irreducibility assumption* for Markov games because the irreducibility assumption implies that there is a positive probability that any state is visited from any state within $|S|$ stages. Furthermore, it reduces to the ergodicity property of Markov decision problems, e.g., see Kearns and Singh [2002], if one of the players has only one action at every state.

As an example, $\alpha_c = c^{-\rho_\alpha}$ and $\beta_c = c^{-\rho_\beta}$, where $0.5 < \rho_\alpha < \rho_\beta \leq 1$ satisfies Assumptions 2-ii and 2'-ii. There exists $\rho \in (0, 2 - 1/\rho_\alpha)$ for the latter since $\rho_\alpha \in (0.5, 1)$. To satisfy Assumption 2-ii, the players can choose the temperature parameter $\{\tau_c\}_{c>0}$ as

$$\tau_c = \frac{1}{c}\bar{\tau} + \left(1 - \frac{1}{c}\right)\epsilon, \quad \forall c > 0 \tag{11}$$

with some $\bar{\tau} > 0$. On the other hand, to satisfy Assumption 2'-ii, they can choose the temperature parameter $\{\tau'_c\}_{c>0}$ as

$$\tau'_c = \bar{\tau}\left(1 + \bar{\tau}\frac{\rho_\alpha \rho}{4D}\log(c)\right)^{-1}, \quad \forall c > 0. \tag{12}$$

Alternative to (11), $\tau_c = \max\{\epsilon, \tau'_c\}$ also satisfies Assumption 2-ii while having similar nature with (12). We provide the relevant technical details in the supplementary material.

We have the following key properties for the estimate sequence generated by our learning dynamics.

**Proposition 1.** *Since $\bar{\alpha}^i_k \in (0,1]$ for all $k \geq 0$, $\beta_c \in (0,1)$ for all $c > 0$, $\|\hat{q}^i_{s,0}\|_\infty \leq D$ and $|\hat{v}^i_{s,0}| \leq D$ for all $(i,s)$, the iterates are bounded, i.e., $\|\hat{q}^i_{s,k}\|_\infty \leq D$ and $|\hat{v}^i_{s,k}| \leq D$ for all $(i,s)$ and $k \geq 0$.*

**Proposition 2.** *Suppose that Assumption 1-i and either Assumption 2-ii or 2'-ii hold. Then, there exists $C_s \in \mathbb{Z}_+$ for each $s \in S$ such that $\alpha_c \exp(2D/\tau_c) < \min_{i=1,2}\{|A^i_s|^{-1}\}$, for all $c \geq C_s$. Correspondingly, the update of the local Q-function estimate (8) reduces to*

$$\hat{q}^i_{s_k,k+1}[a^i_k] = \hat{q}^i_{s_k,k}[a^i_k] + \frac{\alpha_{\#s_k}}{\bar{\pi}^i_k[a^i_k]}\left(r^i_k + \gamma\hat{v}^i_{s_{k+1},k} - \hat{q}^i_{s_k,k}[a^i_k]\right), \tag{13}$$

*for all $\#s_k \geq C_{s_k}$ since $\alpha_{\#s_k}/\bar{\pi}^i_k[a^i_k] \leq |A^i_{s_k}|\alpha_{\#s_k}\exp(2D/\tau_{\#s_k}) \leq 1$ by (7).*

**Proposition 3.** *Suppose that either Assumption 2 or Assumption 2' holds. Then, at any stage $k$, there is a fixed positive probability, e.g., $\underline{p} > 0$, that the game visits any state $s$ at least once within $n$-stages independent of how players play. Therefore, $\#s \to \infty$ as $k \to \infty$ with probability 1.*

Proposition 1 says that defining $\bar{\alpha}^i_k = \min\{1, \alpha_{\#s_k}/\bar{\pi}^i_k[a^i_k]\}$ ensures that the iterates remain bounded. On the other hand, Propositions 2 and 3 say that the update of the local $Q$-function estimates reduces to (13) where $\bar{\alpha}^i_k = \alpha_{\#s_k}/\bar{\pi}^i_k[a^i_k]$ after a finite number of stages, almost surely. The following theorem characterizes the convergence properties of the $Q$-learning dynamics presented.

**Theorem 1.** *Suppose that both players follow the learning dynamics described in Table 1 and Assumption 1 holds. Let $v^i_{\pi_*}$ and $Q^i_{\pi_*}$, as described resp. in (3) and (4), be the unique values associated with some equilibrium profile $\pi_* = (\pi^1_*, \pi^2_*)$ of the underlying zero-sum Markov game. Then, the asymptotic behavior of the value function estimates $\{\hat{v}^i_{s,k}\}_{k\geq 0}$ is given by*

$$\limsup_{k\to\infty}|\hat{v}^i_{s,k} - v^i_{\pi_*}(s)| \leq \epsilon\xi g(\gamma), \quad \text{under Assumption 2,} \tag{14a}$$

$$\lim_{k\to\infty}|\hat{v}^i_{s,k} - v^i_{\pi_*}(s)| = 0, \qquad \text{under Assumption 2',} \tag{14b}$$

*for all $(i,s) \in \{1,2\} \times S$, with probability (w.p.) 1, where $\xi := \max_{s'\in S}\left\{\log(|A^1_{s'}||A^2_{s'}|)\right\}$, and $g(\gamma) = \frac{2+\lambda-\lambda\gamma}{(1-\lambda\gamma)(1-\gamma)}$ with some $\lambda \in (1, 1/\gamma)$.*

*Furthermore, let $\hat{\pi}^i_{s,k}$ be the weighted time-average of the smoothed best response updated as*

$$\hat{\pi}^i_{s,k+1} = \hat{\pi}^i_{s,k} + \mathbf{1}_{\{s=s_k\}}\alpha_{\#s}\left(\bar{\pi}^i_k - \hat{\pi}^i_{s,k}\right).$$

*Then, the asymptotic behavior of these weighted averages $\{\hat{\pi}^i_k\}_{k\geq 0}$ is given by*

$$\limsup_{k\to\infty}\left(\max_{\pi^i}v^i_{\pi^i,\hat{\pi}^{-i}_k}(s) - v^i_{\hat{\pi}^i_k,\hat{\pi}^{-i}_k}(s)\right) \leq \epsilon\xi h(\gamma), \quad \text{under Assumption 2,} \tag{15a}$$

$$\lim_{k\to\infty}\left(\max_{\pi^i}v^i_{\pi^i,\hat{\pi}^{-i}_k}(s) - v^i_{\hat{\pi}^i_k,\hat{\pi}^{-i}_k}(s)\right) = 0, \qquad \text{under Assumption 2',} \tag{15b}$$

*for all $(i,s) \in \{1,2\} \times S$, w.p. 1, where $h(\gamma) = \left[4\gamma \cdot g(\gamma) + 2(1+\lambda)/(1-\lambda\gamma)\right]/(1-\gamma)$, i.e., these weighted-average strategies converge to near or exact equilibrium depending on whether Assumption 2 or 2' hold.*

A brief sketch of the proof is as follows: We decouple the dynamics specific to a single state from others by addressing the asynchronous update of the local $Q$-function estimate and the diminishing temperature parameter. We then approximate the dynamics specific to a single state via its limiting ordinary differential equation (o.d.e.) as if the iterates evolving at the slow timescale are time-invariant. We present a novel Lyapunov function for the limiting o.d.e. to characterize the limit set of the discrete-time update. This Lyapunov function shows that the game perceived by the agents become zero-sum asymptotically and the local $Q$-function estimates are asymptotically belief-based. Finally, we use this limit set characterization to show the convergence of the dynamics across every state by using asynchronous stochastic approximation methods, e.g., see Tsitsiklis [1994].

The following corollary to Theorem 1 highlights the *rationality* property of our learning dynamics.

**Corollary 1.** *Suppose that player $-i$ follows an (asymptotically) stationary strategy $\{\tilde{\pi}_s^{-i} \in \mathrm{int}\Delta(A_s^i)\}_{s\in S}$ while player $i$ adopts the learning dynamics described in Table 1, and Assumption 1 holds. Then, the asymptotic behavior of the value function estimate $\{\hat{v}_{s,k}^i\}_{k\geq 0}$ is given by*

$$\limsup_{k\to\infty} \left| \hat{v}_{s,k}^i - \max_{\pi^i} v_{\pi^i,\tilde{\pi}^{-i}}^i(s) \right| \leq \epsilon \xi^i g(\gamma), \quad \textit{under Assumption 2,} \tag{16a}$$

$$\lim_{k\to\infty} \left| \hat{v}_{s,k}^i - \max_{\pi^i} v_{\pi^i,\tilde{\pi}^{-i}}^i(s) \right| = 0, \quad \textit{under Assumption 2'} \tag{16b}$$

*for all $s \in S$, w.p. 1, where $\xi^i := \max_{s'\in S}\left\{\log(|A_{s'}^i|)\right\}$ and $g(\cdot)$ is as described in Theorem 1.*

*Furthermore, the asymptotic behavior of the weighted averages $\{\hat{\pi}_k^i\}_{k\geq 0}$, described in Theorem 1, is given by*

$$\limsup_{k\to\infty} \left( \max_{\pi^i} v_{\pi^i,\tilde{\pi}^{-i}}^i(s) - v_{\hat{\pi}_k^i,\tilde{\pi}^{-i}}^i(s) \right) \leq \epsilon \xi^i h(\gamma), \quad \textit{under Assumption 2,} \tag{17a}$$

$$\lim_{k\to\infty} \left( \max_{\pi^i} v_{\pi^i,\tilde{\pi}^{-i}}^i(s) - v_{\hat{\pi}_k^i,\tilde{\pi}^{-i}}^i(s) \right) = 0, \quad \textit{under Assumption 2',} \tag{17b}$$

*for all $s \in S$, w.p. 1, where $h(\gamma)$ is as described in Theorem 1, i.e., these weighted-average strategies converge to near or exact best-response strategy, depending on whether Assumption 2 or 2' hold.*

## 4 Simulation Results

All the simulations are executed on a desktop computer equipped with a 3.7 GHz Hexa-Core Intel Core i7-8700K processor with Matlab R2019b. The device also has two 8GB 3000MHz DDR4 memories and a NVIDIA GeForce GTX 1080 8GB GDDR5X graphic card. For illustration, we consider a zero-sum Markov game with 5 states and 3 actions at each state, i.e., $S = \{1, 2, \cdots, 5\}$ and $A_s^i = \{1, 2, 3\}$. The discount factor $\gamma = 0.6$. The reward functions are chosen randomly in a way that $r_s^1(a^1, a^2) \propto \bar{r}_{s,a^1,a^2} \cdot \exp(s^2)$ for $s \in S$, where $\bar{r}_{s,a^1,a^2}$ is uniformly drawn from $[-1, 1]$. $r_s^1(a^1, a^2)$ is then normalized by $\max_{s,a^1,a^2}\{r_s^1(a^1, a^2)\}$ so that $|r_s^i(a^1, a^2)| \leq R = 1$ for all $(i, s, a^1, a^2)$. For the state transition dynamics $p$, we construct two cases, **Case 1** and **Case 2** by randomly generating transition probabilities, so that they satisfy Assumptions 2-i and 2'-i, respectively. For both cases, we choose $\alpha_c = 1/c^{0.9}$ and $\beta_c = 1/c$ with $\rho_\alpha = 0.9$, $\rho_\beta = 1$, and $\rho = 0.7$, and set $\tau_c$ in accordance with (11) and (12), respectively. For **Case 1**, we choose $\epsilon = 2 \times 10^{-4}$ and $\bar{\tau} = 4.5 \times 10^4$; for **Case 2**, we choose $\bar{\tau} = 0.07$. Note that the different choices of $\bar{\tau}$ are due to the different decreasing rates of $\tau_c$ in (11) and (12), and are chosen to generate aesthetic plots. The simulation results are illustrated in Figure 2, which are generated after 20 runs of our learning dynamics.

As shown in Figure 2 (a), for **Case 1**, the value function estimates successfully converge to the neighborhood of the Nash equilibrium values, where the size of the neighborhood is indeed controlled by (14a). For **Case 2**, it is shown in Figure 2 (b) that the value function estimates converge to the Nash equilibrium, as $\tau \to 0$. These observations have corroborated our theory established in §3. Moreover, it is observed that the variance of the iterates decreases as they converge to the (neighborhood of) Nash equilibrium, implying the almost-sure convergence guarantees we have established.

Besides the illustrative example, we have also tested our learning dynamics on larger scale games, and validated our theory for this case (see Figure 3). See §G for more details of the example.

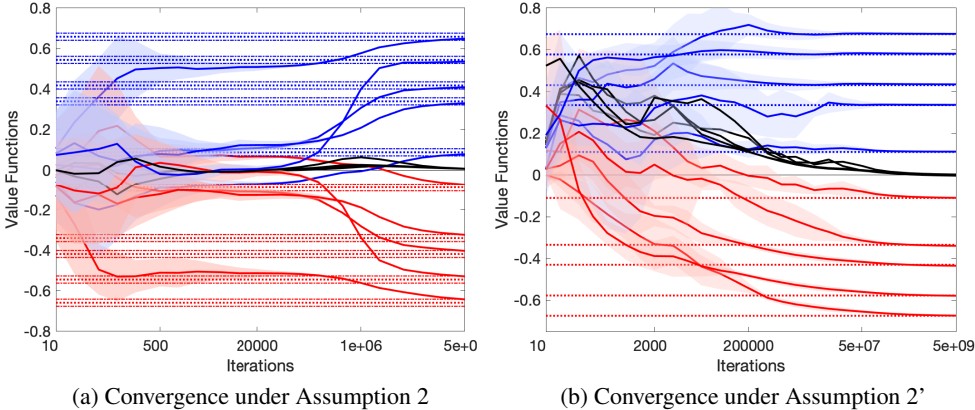

(a) Convergence under Assumption 2      (b) Convergence under Assumption 2'

Figure 2: Convergence of the value function estimates after 20 runs of the decentralized $Q$-learning dynamics. The **red** and **blue** curves represent the quantities for players 1 and 2, respectively. The solid **red**/**blue** lines and the bar-areas around them denote the average value function estimates $\{\hat{v}^i_{s,k}\}_{k\geq 0}$ at all states $s \in S$ and the standard deviations, after the 20 runs, respectively. The solid **black** lines denote the summation of the estimates $\{\hat{v}^1_{s,k} + \hat{v}^2_{s,k}\}_{k\geq 0}$ at each $s$ (which should converge to (almost) zero asymptotically). The dotted lines ‑‑‑‑‑/‑‑‑‑ denote the actual Nash equilibrium values; the dashed-dotted lines ‑‑‑‑‑/‑‑‑‑ denote the boundaries of the neighborhoods around the Nash equilibrium given in Theorem 1. (a) Convergence to the neighborhood of the Nash equilibrium value, under Assumption 2. (b) Convergence to the Nash equilibrium value, under Assumption 2'.

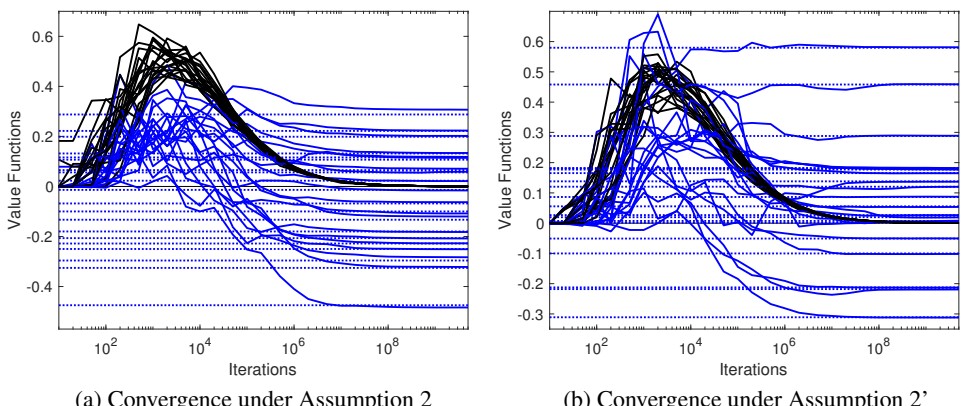

(a) Convergence under Assumption 2      (b) Convergence under Assumption 2'

Figure 3: Convergence of the value function estimates of the decentralized $Q$-learning dynamics. The **blue** curves represents the quantities for player 2. The solid **blue** lines denote the value function estimates $\{\hat{v}^2_{s,k}\}_{k\geq 0}$ at all states $s \in S$. The solid **black** lines denote the summation of the estimates $\{\hat{v}^1_{s,k} + \hat{v}^2_{s,k}\}_{k\geq 0}$ at each $s$ (which should converge to (almost) zero asymptotically). The dotted lines ‑‑‑‑‑ denote the actual Nash equilibrium values. (a) Convergence to the neighborhood of the Nash equilibrium value, under Assumption 2. (b) Convergence to the Nash equilibrium value, under Assumption 2'.

## 5   Concluding Remarks

This paper has studied decentralized multi-agent reinforcement learning in zero-sum Markov games. We have developed a decentralized $Q$-learning dynamics with provable convergence guarantees to the (neighborhoods of the) Nash equilibrium value of the game. Unlike many existing MARL algorithms, our learning dynamics is both rational and convergent, and only based on the payoffs received and the local actions executed by each agent. It also requires neither asymmetric stepsizes/update rules or any coordination for the agents, nor even being aware of the existence of the opponent.

## Acknowledgments and Disclosure of Funding

M. O. Sayin was with the Laboratory for Information and Decision Systems at MIT when this paper was submitted. K. Zhang and A. Ozdaglar were supported by DSTA grant 031017-00016. T. Başar was supported in part by ONR MURI Grant N00014-16-1-2710 and in part by AFOSR Grant FA9550-19-1-0353.

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
