# Supplementary Materials for "Decentralized $Q$-Learning in Zero-sum Markov Games"

## A  Related Work

We here focus on the related works on MARL with provable convergence guarantees.

### A.1  Multi-agent RL in Markov Games

Stemming from the seminal work Littman [1994], Markov games have been widely recognized as the benchmark setting for MARL. Littman [1994] focused on the zero-sum setting, and developed minimax $Q$-learning algorithm with asymptotic converge guarantees [Szepesvári and Littman, 1999]. However, this algorithm requires each agent to observe the opponent's action. More importantly, each agent is fully aware of the zero-sum game being played, and solves a linear program to solve a matrix game at each iteration. Subsequently, Bowling and Veloso [2001] proposed that a preferable MARL algorithm should be both *rational* and *convergent*: a rational algorithm ensures that the iterates converge to the opponent's best-response if the opponent converges to a stationary policy; while a convergent algorithm ensures convergence to some equilibrium if all the agents apply the learning dynamics. In this sense, minimax $Q$-learning is not rational. In contrast, our learning dynamics is both rational and convergent.

In the same vein as minimax $Q$-learning, with coordination among agents, asymptotic convergence has also been established for other $Q$-learning variants beyond the zero-sum setting [Littman, 2001, Hu and Wellman, 2003, Greenwald et al., 2003]. Borkar [2002] has also established the asymptotic convergence of an actor-critic algorithm to a weaker notion of generalized Nash equilibrium. Recently, there is an increasing interest in studying the *non-asymptotic* performance of MARL in Markov games [Pérolat et al., 2015, Wei et al., 2017, Sidford et al., 2019, Xie et al., 2020, Bai and Jin, 2020, Bai et al., 2020, Zhang et al., 2019, 2020, Shah et al., 2020, Liu et al., 2020, Zhao et al., 2021]. These algorithms are in essence centralized, in that they require either the control of both agents [Pérolat et al., 2015, Sidford et al., 2019, Xie et al., 2020, Bai and Jin, 2020, Bai et al., 2020, Zhang et al., 2019, 2020, Shah et al., 2020, Liu et al., 2020, Zhao et al., 2021], or at least the observation of the opponent's actions [Wei et al., 2017, Xie et al., 2020].

Two closely related recent papers Leslie et al. [2020] and Sayin et al. [2020] have presented, respectively, continuous-time best response dynamics and discrete-time fictitious play dynamics that can converge to an equilibrium in zero-sum Markov games. They have established provable convergence by addressing the non-stationarity issue through a two-timescale framework. Though these two-timescale dynamics share a similar flavor with our approach, still, observing the opponent's mixed strategy (in Leslie et al. [2020]) or actions (in Sayin et al. [2020]) is indispensable in them and plays an important role in their analysis. This is in stark contrast to our dynamics that require minimal information, i.e., being radically uncoupled [Foster and Young, 2006, Leslie and Collins, 2005].

### A.2  Decentralized Multi-agent Learning

Decentralized learning is a desired property, and has been studied for matrix games (single-state Markov games) under the framework of *no-regret learning* [Cesa-Bianchi and Lugosi, 2006, Freund and Schapire, 1999, Mertikopoulos and Zhou, 2019]. Leslie and Collins [2005] also proposed individual soft $Q$-learning dynamics for zero-sum matrix games. For general Markov games, however, it is known that blindly applying independent/decentralized $Q$-learning can easily diverge, due to the *non-stationarity* of the environment [Tan, 1993, Boutilier, 1996, Matignon et al., 2012]. Despite this, the decentralized paradigm has still attracted continuing research interest [Arslan and Yuksel, 2017, Pérolat et al., 2018, Daskalakis et al., 2020, Tian et al., 2020, Wei et al., 2021], since it is much more scalable and natural for agents to implement. Notably, these works are not as decentralized and as general as our learning dynamics.

Specifically, the algorithm in Arslan and Yuksel [2017] requires the agents to coordinately explore every multiple iterations (the exploration phase), without changing their policies within each exploration phase, in order to create a stationary environment for each agent. Similar to our work, Pérolat et al.

[2018] also proposed decentralized and two-timescale algorithms, which, however, is an actor-critic algorithm where the value functions are estimated at a faster timescale (critic step), and the policy is improved at a slower one (actor step). More importantly, the algorithm only applies to Markov games with a "multistage" structure, in which each state can only be visited once. Establishing convergence in general zero-sum Markov games is posted as an open problem in Pérolat et al. [2018]. In Daskalakis et al. [2020], the agents have to coordinate to use two-timescale stepsizes in the updates. In contrast, our learning dynamics does not require any coordination among agents, and each agent plays a symmetric role in learning, referred to as *strongly independent* in Daskalakis et al. [2020]. In fact, developing provable guarantees for strongly independent algorithms is considered as an important open question in Daskalakis et al. [2020].

Two recent works Tian et al. [2020], Wei et al. [2021] studied the decentralized setting that is closest to ours. Tian et al. [2020] focused on the exploration aspect for finite-horizon settings, and considered a weak notion of regret. It is unclear if the learning dynamics converge to any equilibrium when both agents apply it[4]. Contemporaneously, Wei et al. [2021] presented an interesting optimistic variant of the gradient descent-ascent method, with a strong guarantee of last-iterate convergence rates, which shares all the desired properties as our learning dynamics. The algorithm is delicately designed and different from the common value/policy-based RL update rules, e.g., $Q$-learning, as in our work. Moreover, to characterize finite-time convergence, in the model-free setting, the agents need to coordinate to interact multiple steps at each iteration of the algorithm, while our learning dynamics is coordination-free with natural update rules. These two works can thus be viewed as orthogonal to ours.

After submitting the first draft of our paper, we were reminded of an independent and concurrent work of Guo et al. [2021], which also studied a decentralized learning setting in zero-sum Markov games. We summarize the substantial differences between the two works as follows.

**Motivation:** In Guo et al. [2021], being "decentralized" is defined as "each player not knowing the opponent's action", to "protect the privacy", and the goal is to "compute" the Nash equilibrium of the game; in contrast, in our work, in addition to "being oblivious to the opponent's action", we also allow no "coordination" among agents, so that each agent can simply run the learning dynamics "individually", without even being aware of the existence of the opponent. The agents in our setting are considered as self-interested decision-makers, who seek to adapt to the opponent's play by inferring it from the rewards received without seeing the opponent's actions. The Nash equilibrium, on the other hand, is the result that "emerge" naturally when both agents follow this self-interested learning dynamics (and we have proved this). Finally, as our learning dynamics are oblivious to the opponent and are adaptive to the opponent, we expect it to converge beyond the zero-sum setting (e.g., the identical-interest setting), which is one of our ongoing research directions. In contrast, the algorithm in Guo et al. [2021] is specifically developed for the zero-sum setting. These motivations differ fundamentally from Guo et al. [2021] (and thus creates very different technical challenges, as detailed below).

**Learning dynamics (Algorithms):** The algorithm in Guo et al. [2021], is actor-critic, which is a type of policy-based RL method; the learning dynamics in our work is Q-learning based, which belongs to value-based RL methods. More importantly, the update-rule in Guo et al. [2021], is of "double-loop" form, in the sense that it fixes the iterate of Player 1 while updating Player 2's policy, so that a "best-response" policy of Player 2 can be obtained. This is an asymmetric update-rule, and requires coordination between agents. In contrast, our learning dynamics are "symmetric", without such a double-loop coordination, where each agent simply runs her own $Q$-learning dynamics.

**Assumptions and results:** Guo et al. [2021] considers a function approximation setting, and assumes that: 1) the "double-loop" update can be implemented by the agents in the decentralized setting; 2) the concentration (or "Concentrability") coefficient is finite (Assumption 4.1), for "an arbitrary sequence of policies"; 3) samples are drawn i.i.d. from the stationary state-action distribution; 4) projection of the iterates onto some ball with radius $R$, to ensure the iterates' stability; and 5) zero approximation error of the Bellman operator (Assumption 4.2). Under these assumptions, non-asymptotic convergence results were established. In contrast, our work considers a fundamental tabular setting, and without making these assumptions (1-4), with instead asymptotic convergence

---

[4]Note that the same update rule with different stepsize and bonus choices and a certified policy technique, however, can return a non-Markovian approximate Nash equilibrium policy pair in the self-play setting, by storing the whole history of the learning process; see Bai et al. [2020] for more details.

guarantees. With these significantly different assumptions, it is not clear if one paper's result implies the other's.

**Analysis techniques (Technical novelty):** The analyses, as well as the technical novelties in both papers are not comparable. The analysis technique in Guo et al. [2021], is a mirror-descent type of analysis, based on the convergence analysis of policy gradient (and actor-critic) algorithms in single-agent RL. The techniques in our paper, however, are based on stochastic approximation theory, a classic technique in showing the convergence of $Q$-learning. The challenges we need to address (our technical novelties) mainly lie in constructing a Lyapunov function and stability of the iterates, within this non-standard two-timescale stochastic approximation setting, with asynchronous updates. Such challenges would not be encountered in the analysis of Guo et al. [2021], making the technical novelties of the two papers fundamentally different.

# B Examples

In this section, we provide three sets of parameter examples and highlight whether they satisfy Assumptions 1 and 2-ii or Assumptions 1 and 2'-ii. Recall that Assumptions 2-i and 2'-i do not impose conditions on the step sizes nor the temperature parameter.

**Example 1.** *Set the step sizes as $\alpha_c = c^{-\rho_\alpha}$, $\beta_c = c^{-\rho_\beta}$, where $1/2 < \rho_\alpha < \rho_\beta \leq 1$ and the temperature parameter as*

$$\tau_c = \frac{1}{c}\bar{\tau} + \left(1 - \frac{1}{c}\right)\epsilon \tag{18}$$

*for some $\bar{\tau} > 0$.*

In the following, we show that Example 1 satisfies Assumption 1 and 2-ii. Assumption 1-i holds since $\sum_{c>0}(1/c)^\rho$ is convergent if $\rho > 1$, and divergent if $\rho \leq 1$. Assumption 1-ii holds since there exists a non-decreasing polynomial, e.g., $C(x) = M_o x^m$, for all $x \geq 1$, where $M_o := M^{-\rho_\beta/(\rho_\beta - \rho_\alpha)} > 0$ and $m \in \{m' \in \mathbb{Z}_+ : m' \geq 1/(\rho_\beta - \rho_\alpha)\}$. Particularly, we have

$$\frac{\beta_l}{\alpha_c} = \frac{c^{\rho_\alpha}}{l^{\rho_\beta}} > \lambda \quad \Leftrightarrow \quad l < \left(\frac{1}{\lambda}\right)^{\frac{1}{\rho_\beta}} c^{\frac{\rho_\alpha}{\rho_\beta}}. \tag{19}$$

We claim that $Mc \geq \lambda^{-1/\rho_\beta} c^{\rho_\alpha/\rho_\beta}$ for all $c \geq C(\lambda^{-1})$ because $\lambda^{-1/\rho_\beta} c^{\rho_\alpha/\rho_\beta} \leq Mc$ yields

$$c^{1-\frac{\rho_\alpha}{\rho_\beta}} \geq \left(\frac{1}{\lambda}\right)^{\frac{1}{\rho_\beta}} \frac{1}{M} \quad \Leftrightarrow \quad c \geq \left(\frac{1}{\lambda}\right)^{\frac{1}{\rho_\beta - \rho_\alpha}} \left(\frac{1}{M}\right)^{\frac{\rho_\beta}{\rho_\beta - \rho_\alpha}} = M_o \left(\frac{1}{\lambda}\right)^{\frac{1}{\rho_\beta - \rho_\alpha}}. \tag{20}$$

On the other hand, we have $C(\lambda^{-1}) = M_o \lambda^{-m} \geq M_o \lambda^{-1/(\rho_\beta - \rho_\alpha)}$ for all $\lambda \in (0,1)$ since $m \geq 1/(\rho_\beta - \rho_\alpha)$ by its definition.

Assumption 2-ii holds since $\{\tau_c\}$ monotonically decreases to $\epsilon$ as $c \to \infty$, and

$$\frac{\tau_{c+1} - \tau_c}{\alpha_c} = c^{\rho_\alpha} \left[\frac{1}{c+1}\bar{\tau} + \left(1 - \frac{1}{c+1}\right)\epsilon - \frac{1}{c}\bar{\tau} - \left(1 - \frac{1}{c}\right)\epsilon\right] \tag{21}$$

$$= \frac{c^{\rho_\alpha}}{c(c+1)}(\epsilon - \bar{\tau}), \tag{22}$$

which goes to zero as $c \to \infty$ since $\rho_\alpha < 1$, and $\sum_{c>0} c^{-2\rho_\alpha} < \infty$ since $2\rho_\alpha > 1$. $\square$

**Example 2.** *Set the step sizes as $\alpha_c = c^{-\rho_\alpha}$, $\beta_c = c^{-\rho_\beta}$, where $1/2 < \rho_\alpha < \rho_\beta \leq 1$ and the temperature parameter as*

$$\tau'_c = \bar{\tau}\left(1 + \bar{\tau}\frac{\rho_\alpha \rho}{4D}\log(c)\right)^{-1} \tag{23}$$

*for some $\rho \in (0, 2 - 1/\rho_\alpha)$ and $\bar{\tau} > 0$.*

In the following, we show that Example 2 satisfies Assumption 1 and 2'-ii. Example 2 shares the same step sizes with Example 1. Therefore, Assumption 1 holds as shown above for Example 1. On the other hand, Assumption 2'-ii also holds since $\{\tau'_c\}$ monotonically decreases to 0 as $c \to \infty$, and

$$0 \geq \frac{\tau'_{c+1} - \tau'_c}{\alpha_c} \geq \frac{c^{\rho_\alpha}(\log(c+1) - \log(c))}{\log(c+1)\log(c)}, \tag{24}$$

where the right-hand side goes to zero as $c \to \infty$, and

$$\alpha_c^\rho \exp\left(\frac{4D}{\tau_c'}\right) = c^{-\rho_\alpha \rho} \exp\left(\frac{4D}{\bar{\tau}}\left(1 + \bar{\tau}\frac{\rho_\alpha \rho}{4D}\log(c)\right)\right) \tag{25}$$

$$= c^{-\rho_\alpha \rho} \exp\left(\frac{4D}{\bar{\tau}}\right) \exp\left(\rho_\alpha \rho \log(c)\right) \tag{26}$$

$$= \exp\left(\frac{4D}{\bar{\tau}}\right), \quad \forall c > 0, \tag{27}$$

which implies that $\alpha_c^\rho \exp(4D/\tau_c') \leq C'$ for all $c > 0$ when $C' = \exp(4D/\bar{\tau})$. $\qquad \square$

**Example 3.** *Set the step sizes as $\alpha_c = c^{-\rho_\alpha}$, $\beta_c = c^{-\rho_\beta}$, where $1/2 < \rho_\alpha < \rho_\beta \leq 1$ and the temperature parameter as $\tau_c = \max\{\epsilon, \tau_c'\}$, where $\tau_c'$ is as described in (23).*

In the following, we show that Example 1 satisfies Assumption 1 and 2-ii. Example 3 shares the same step sizes with Example 1. Therefore, Assumption 1 holds as shown above for Example 1. On the other hand, Assumption 2-ii also holds since $\{\tau_c\}$ monotonically decreases to $\epsilon$ as $c \to \infty$ (which follows since $\{\tau_c'\}$ monotonically decreases to 0 as $c \to \infty$), and we again have the inequality (24) and $\sum_{c>0} c^{-2\rho_\alpha} < \infty$ since $2\rho_\alpha > 1$. $\qquad \square$

## C   Proofs of Propositions 1-3

**Proposition 1**.   *Since $\bar{\alpha}_k^i \in (0,1]$ for all $k \geq 0$, $\beta_c \in (0,1)$ for all $c > 0$, $\|\hat{q}_{s,0}^i\|_\infty \leq D$ and $|\hat{v}_{s,0}^i| \leq D$ for all $(i,s)$, the iterates are bounded, e.g., $\|\hat{q}_{s,k}^i\|_\infty \leq D$ and $|\hat{v}_{s,k}^i| \leq D$ for all $(i,s)$ and $k \geq 0$.*

*Proof:*   The proof follows from the fact that the initial iterates are picked within the compact set and they continue to remain inside it since they are always updated to a convex combination of two points inside. $\qquad \square$

**Proposition 2**.   *Suppose that Assumption 1-i and either Assumption 2-ii or 2'-ii hold. Then, there exists $C_s \in \mathbb{Z}_+$ for each $s \in S$ such that $\alpha_c \exp\left(2D/\tau_c\right) < \min_{i=1,2}\{|A_s^i|^{-1}\}$, for all $c \geq C_s$. Correspondingly, the update of the local Q-function estimate (8) reduces to*

$$\hat{q}_{s_k,k+1}^i[a_k^i] = \hat{q}_{s_k,k}^i[a_k^i] + \frac{\alpha_{\#s_k}}{\bar{\pi}_k^i[a_k^i]}\left(r_k^i + \gamma\hat{v}_{s_{k+1},k}^i - \hat{q}_{s_k,k}^i[a_k^i]\right),$$

*for all $\#s_k \geq C_{s_k}$ since $\alpha_{\#s_k}/\bar{\pi}_k^i[a_k^i] \leq |A_{s_k}^i|\alpha_{\#s_k}\exp\left(2D/\tau_{\#s_k}\right) \leq 1$ by (7).*

*Proof:*   If Assumption 2-ii holds, then $\alpha_c \exp(2D/\tau_c) \leq \alpha_c \exp(2D/\epsilon)$. Since $\alpha_c \to 0$ as $c \to \infty$ by Assumption 1-i, there exists such $C_s$.

If Assumption 2'-ii holds, then

$$\alpha_c \exp\left(\frac{2D}{\tau_c}\right) = \alpha_c^{1-\rho/2}\left(\alpha^\rho \exp\left(\frac{4D}{\tau_c}\right)\right)^{1/2} \tag{28}$$

$$\leq \alpha_c^{1-\rho/2}\sqrt{C'}, \quad \forall c \geq C. \tag{29}$$

Since $1 - \rho/2 > 0$ and $\alpha_c \to 0$ as $c \to \infty$, there exists such $C_s$. $\qquad \square$

**Proposition 3**.   *Suppose that either Assumption 2 or Assumption 2' holds. Then, at any stage $k$, there is a fixed positive probability, e.g., $\underline{p} > 0$, that the game visits any state $s$ at least once within $n$-stages independent of how players play. Therefore, $\#s \to \infty$ as $k \to \infty$ with probability 1.*

*Proof:*   By Borel-Cantelli Lemma, if we have

$$\sum_{k \geq 0} \mathbb{P}\{\#_k s \leq \lambda\} < \infty, \quad \forall \lambda \in \mathbb{N}, \tag{30}$$

then we have $\#_k s \to \infty$ as $k \to \infty$ ith probability 1. To show (30), we partition the time axis into $n$-stage intervals and introduce an auxiliary counting process $\bar{\#}_k^n s$ that increases by 1 at the end of each interval if state $s$ is visited at least once within the last $n$-stages. By its definition, we have

$\bar{\bar{\#}}_k^n s \leq \#_k s$. Correspondingly, we have $\mathbb{P}\{\#_k s \leq \lambda\} \leq \mathbb{P}\{\bar{\bar{\#}}_k^n s \leq \lambda\}$. The right-hand side is one for all $k < n\lambda$. On the other hand, for $k \geq n\lambda$, we have

$$\mathbb{P}\{\bar{\bar{\#}}_k^n s \leq \lambda\} \leq \sum_{l=0}^{\lambda} \binom{\lfloor \frac{k}{n} \rfloor}{l} 1^l (1-\underline{p})^{\lfloor \frac{k}{n} \rfloor - l} \tag{31}$$

$$\leq (1-\underline{p})^{\lfloor \frac{k}{n} \rfloor - \lambda} \sum_{l=0}^{\lambda} \binom{\lfloor \frac{k}{n} \rfloor}{l} \tag{32}$$

since $(1-\underline{p}) < 1$.

Next, we can resort to the following inequality [Flum and Grohe, 2006, Lemma 16.19]

$$\sum_{l=0}^{L} \binom{k}{l} \leq 2^{H_2(L/k)k}, \quad \text{if } L/k \leq 1/2, \tag{33}$$

where $H_2(p) := -p \log(p) - (1-p)\log(1-p)$. Therefore, for $k \geq 2n\lambda$, (32) and (33) yield that

$$\mathbb{P}\{\bar{\bar{\#}}_k^n s \leq \lambda\} \leq \frac{1}{(1-\underline{p})^\lambda} \left[ (1-\underline{p}) 2^{H_2(\lambda/\lfloor k/n \rfloor)} \right]^{\lfloor k/n \rfloor}. \tag{34}$$

Since $H_2(p) \geq 0$ is an increasing continous function for $p \in (0, 0.5)$ and $H_2(0) = 0$, there exists $\kappa \in \mathbb{N}$ such that $(1-\underline{p})2^{H_2(\lambda/\lfloor k/n \rfloor)} < (1-\underline{p})2^{H_2(\lambda/\lfloor \kappa/n \rfloor)} < 1$ for all $k \geq \kappa$. Define $\xi := (1-\underline{p})2^{H_2(\lambda/\lfloor \kappa/n \rfloor)}$. Then we have

$$\sum_{k \geq 0} \mathbb{P}\{\#_k s \leq \lambda\} \leq \kappa + \frac{1}{(1-\underline{p})^\lambda} \sum_{k=\kappa+1}^{\infty} \xi^{\lfloor k/n \rfloor} \leq \kappa + \frac{1}{\xi(1-\underline{p})^\lambda} \sum_{k=\kappa+1}^{\infty} (\xi^{1/n})^k \tag{35}$$

and the right-hand side is convergent since $\xi^{1/n} < 1$, which completes the proof. $\square$

# D    Preliminary Information on Stochastic Approximation Theory

Here, we present two preliminary results. The former uses a continuous-time approximation to analyze a discrete-time update [Benaim, 1999]. The latter is about characterizing the convergence properties of an asynchronous discrete-time update by exploiting certain bounds on their evolution [Sayin et al., 2020].

## D.1    Stochastic Approximation via Lyapunov Function

The following theorem (follows from [Benaim, 1999, Proposition 4.1 and Corollary 6.6]) characterizes the conditions sufficient to characterize the convergence properties of a discrete-time update:

$$x_{k+1} = x_k + \lambda_k \left[ F(x_k) + \epsilon_k + \omega_k \right], \tag{36}$$

through its limiting ordinary differential equation (o.d.e.):

$$\frac{dx(t)}{dt} = F(x(t)). \tag{37}$$

**Theorem 2.** *Suppose that there exists a Lyapunov function $V : \mathbb{R}^m \to [0, \infty)$ for (37).[5] Furthermore,*

i) *The step sizes $\{\lambda_k \in [0, 1]\}_{k=0}^{\infty}$ decrease at a suitable rate:*

$$\sum_{k=0}^{\infty} \lambda_k = \infty \text{ and } \sum_{k=0}^{\infty} \lambda_k^2 < \infty. \tag{38}$$

ii) *The iterates $x_k \in \mathbb{R}^m$, for $k = 0, 1, \ldots$, are bounded, e.g., $\sup_k \|x_k\|_\infty < \infty$.*

---

[5]We say that a continous function $V$ is a *Lyapunov function* for a flow provided that for any trajectory of the flow, e.g., $x(t)$, $V(x(t')) < V(x(t))$ for all $t' > t$ if $V(x(t)) > 0$ else $V(x(t')) = 0$ for all $t' > t$.

*iii) The vector field $F : \mathbb{R}^m \to \mathbb{R}^m$ is globally Lipschitz continous.*

*iv) The stochastic approximation term $\omega_k \in \mathbb{R}^m$ satisfies the following condition for all $K > 0$,[6]*

$$\lim_{k \to \infty} \sup_{n > k : \sum_{l=k}^{n-1} \lambda_l \leq K} \left\{ \left\| \sum_{l=k}^{n-1} \lambda_l \omega_l \right\| \right\} = 0. \tag{39}$$

*v) The error term $\epsilon_k \in \mathbb{R}^m$ is asymptotically negligible, i.e., $\lim_{k \to \infty} \|\epsilon_k\| = 0$, with probability 1.*

*Then the limit set of* (36) *is contained in the set*

$$\{x \in \mathbb{R}^m : V(x) = 0\}, \tag{40}$$

*with probability 1.*

### D.2 Asynchronous Stochastic Approximation

Consider the scenarios where we update only a subset of entries of the iterate $x_k$ with specific step sizes. For example, the $l$th entry of the iterate $x_k \in \mathbb{R}^m$, denoted by $x_k[l]$, gets updated only at certain (possibly random) time instances with a specific step size $\lambda_{k,l} \in [0, 1]$. Furthermore, there is not necessarily a time-invariant vector field $F$ as in Theorem 2. The following theorem, [Sayin et al., 2020, Theorem 3], characterizes the limit set of $\{x_k\}_{k \geq 0}$ provided that it evolves within a shrinking envelope.[7]

**Theorem 3.** *Suppose that the evolution of $\{x_k\}_{k \geq 0}$ always satisfies the following upper and lower bounds:*

$$x_{k+1}[l] \leq (1 - \lambda_{k,l}) x_k[l] + \lambda_{k,l} (\gamma \|x_k\|_\infty + \bar{\epsilon}_k), \tag{41a}$$

$$x_{k+1}[l] \geq (1 - \lambda_{k,l}) x_k[l] + \lambda_{k,l} (-\gamma \|x_k\|_\infty + \underline{\epsilon}_k), \tag{41b}$$

*where $\gamma \in (0, 1)$ is a discount factor, $\|x_k\|_\infty \leq D$ for all $k \geq 0$ for a fixed $D$, and the specific step sizes satisfy the usual conditions:*

$$\sum_{k=0}^\infty \lambda_{l,k} = \infty \quad and \quad \sum_{k=0}^\infty \lambda_{k,l}^2 < \infty,$$

*and the errors $\bar{\epsilon}_k, \underline{\epsilon}_k \in \mathbb{R}$ satisfy*

$$\limsup_{k \to \infty} |\bar{\epsilon}_k| \leq c \quad and \quad \limsup_{k \to \infty} |\underline{\epsilon}_k| \leq c, \tag{42}$$

*for some $c \geq 0$, with probability 1. Then, we have*

$$\limsup_{k \to \infty} \|x_k\|_\infty \leq \frac{c}{1 - \gamma}, \tag{43}$$

*with probability 1.*

## E  Convergence Analysis: Proof of Theorem 1

The proof is built on the following observation: The update of the value function estimate, (9), can be written as

$$\hat{v}_{s,k+1}^i = \hat{v}_{s,k}^i + \mathbf{1}_{\{s=s_k\}} \beta_{\#s} \left[ T^i(\{\hat{v}_{s',k}^i\}_{s' \in S})[s] - \hat{v}_{s,k}^i + \epsilon_{s,k}^i \right], \tag{44}$$

where the tracking error $\epsilon_{s,k}^i$ is defined by

$$\boxed{\epsilon_{s,k}^i := \bar{\pi}_{s,k}^i \cdot \hat{q}_{s,k}^i - \mathrm{val}^i(\hat{Q}_{s,k}^i)}, \tag{45}$$

---

[6]This is a more general condition than assuming that $\{\omega_k\}$ is a square-integrable Martingale difference sequence, e.g., see [Borkar, 2008, Section 2].

[7]This theorem is a rather straight-forward modification of [Tsitsiklis, 1994, Theorem 3].

and the operator $T^i : \mathbb{R}^{|S|} \to \mathbb{R}^{|S|}$ is defined by

$$T^i(\{\hat{v}^i_{s',k}\}_{s' \in S})[s] := \text{val}^i(\hat{Q}^i_{s,k}), \tag{46}$$

where $\hat{Q}^i_{s,k} \in [-D, D]^{|A^i_s| \times |A^{-i}_s|}$ corresponding to the global $Q$-function is defined by

$$\hat{Q}^i_{s,k}[a^i, a^{-i}] := r^i_s(a^1, a^2) + \gamma \sum_{s' \in S} p(s'|s, a^1, a^2)\hat{v}^i_{s',k}, \quad \forall i = 1, 2, \tag{47}$$

and $\text{val}^i : [-D, D]^{|A^i_s| \times |A^{-i}_s|} \to \mathbb{R}$ is[8] defined by

$$\text{val}^i(Q^i_{s,k}) := \max_{\mu^i \in \Delta(A^i_s)} \min_{\mu^{-i} \in \Delta(A^{-i}_s)} \left\{ (\mu^i)^T Q^i_{s,k} \mu^{-i} \right\}. \tag{48}$$

The non-expansiveness property of $\text{val}^i(\cdot)$, as shown by Shapley [1953], and the discount factor $\gamma \in (0, 1)$ yield that the operator is a contraction, e.g.,

$$\|T^i(v^i) - T^i(\tilde{v}^i)\|_\infty \le \gamma \max_{s \in S} |v^i_s - \tilde{v}^i_s|. \tag{49}$$

Denote the unique fixed point of the contraction $T^i$ by $\{v^i_{s,*}\}_{s \in S}$. Then, we have $v^i_{s,*} = T^i(\{v^i_{s',*}\}_{s' \in S})[s]$. Therefore, the update (44) can be written as

$$\hat{v}^i_{s,k+1} - v^i_{s,*} = \hat{v}^i_{s,k} - v^i_{s,*}$$
$$+ \mathbf{1}_{\{s=s_k\}} \beta_{\#s} \left[ T^i(\{\hat{v}^i_{s',k}\}_{s' \in S})[s] - T^i(\{v^i_{s',*}\}_{s' \in S})[s] - (\hat{v}^i_{s,k} - v^i_{s,*}) + \epsilon^i_{s,k} \right], \tag{50}$$

Based on Proposition 3, Theorem 3 and (44) yield that the asymptotic behavior of the value function estimates can be characterized as follows:

$$\limsup_{k \to \infty} |\hat{v}^i_{s,k} - v^i_{s,*}| \le \frac{1}{1-\gamma} \limsup_{k \to \infty} \epsilon^i_{s,k} \tag{51}$$

for all $(i, s) \in \{1, 2\} \times S$. The rest of the proof is about characterizing the asymptotic behavior of the tracking error (45) and showing that

$$\limsup_{k \to \infty} \epsilon^i_{s,k} \le \frac{1}{1-\gamma} \left( \frac{2 + \lambda - \lambda\gamma}{1 - \lambda\gamma} \right) \max_{s' \in S} \{\log(|A^1_{s'}||A^2_{s'}|)\} \lim_{c \to \infty} \tau_c, \tag{52}$$

for some $\lambda \in (1, 1/\gamma)$.

It is instructive to discuss why the existing results cannot directly address this tracking error's asymptotic behavior. For example, there exist several well-established results on convergence properties of learning dynamics in strategic-form games with repeated play for both zero-sum and potential games, e.g., see Fudenberg and Levine [2009]. The challenge raises since $\hat{Q}^i_{s,k}$ is not time-invariant and it depends on both players' strategies. On the other hand, the existing results to characterize the convergence properties of the classical (single-agent) $Q$-learning is helpful only to obtain (51) and do not address the tracking error (45). Note that if the players are coordinated to play the equilibrium behavior, e.g., as in Shapley's value iteration [Shapley, 1953] or Minimax-Q [Littman, 1994], then the tracking error would be zero by the nature of the updates. However, this would imply that the players are *coordinated* to play the equilibrium since

- Players need to know the zero-sum structure of the game,
- Players always play the conservative strategy against the worst-case strategy of the opponent and do not attempt to take the best reaction when the opponent is not playing the equilibrium strategy,
- Players need to observe the opponent's actions to be able to compute the global $Q$-function associated with the joint actions.

A two-timescale learning dynamics can address the dependence of the $Q$-function estimate on the strategies and correspondingly address the tracking error. However, there are several challenges especially for radically uncoupled schemes, where players do not observe the opponent's actions:

---

[8]Note that $\text{val}^i$ technically also depends on $s$, since $A_s$ depends on $s$. We omit $s$ for notational convenience, and it shall not cause any confusion from the context.

*i)* The local $Q$-function estimates for different state and local action pairs can get updated at different frequencies, which poses a challenge for the two-timescale framework to decouple the dynamics at fast and slow timescales. Particularly, the normalization of the step size in the update of the local $Q$-function estimate can ensure that the estimate for each local action gets updated at the same rate in the expectation. However, this is not sufficient since estimates for some local actions can lag behind even the iterates evolving on the slow timescale.

*ii)* The $Q$-function estimates may not necessarily sum to zero in general when the players keep track of it independently, i.e., if there is no central coordinator providing it to them. This is important because uncoupled learning dynamics cannot converge to an equilibrium in every class of games, as shown in Hart and Mas-Colell [2003].

*iii)* The players can keep track of only local $Q$-function since they cannot observe the opponent's action. However, there may not even exist an opponent (mixed) strategy that can lead to the local $Q$-function estimate, i.e., they may not be belief-based, whereas this is not the case if players can observe the opponent's actions to form a belief on the opponent's strategy.

In the following, we follow a three-step approach to address these challenges:

1. Decoupling dynamics at the fast timescale by addressing Challenge *i)*.
2. Zooming into the local dynamics (i.e., learning dynamics specific to a single state) at the fast timescale to address Challenges *ii)* and *iii)* via a novel Lyapunov function.
3. Zooming out to the global dynamics (i.e., learning dynamcis across every state) at the slow timescale to characterize the asymptotic behavior of the tracking error (45).

In the following, we delve into the details of these steps.

### E.1 Decoupling dynamics at the fast timescale

Distinct to the radically uncoupled settings, the players can update only the local $Q$-function estimate's entry specific to the current local action. Although this is an asynchronous update, the normalization makes the evolution of every entry synchronous in the expectation [Leslie and Collins, 2005]. To show this, we introduce the stochastic approximation error:

$$\omega^1_{s_k,k}[a^1] := \mathbf{1}_{\{a^1_k=a^1\}} \frac{r^1_s(a^1, a^2_k) + \gamma \hat{v}^1_{s_{k+1},k} - \hat{q}^1_{s_k,k}[a^1]}{\overline{\pi}^1_k[a^1]}$$
$$- \mathbb{E}\left\{ \mathbf{1}_{\{a^1_k=a^1\}} \frac{r^1_s(a^1, a^2_k) + \gamma \hat{v}^1_{s_{k+1},k} - \hat{q}^1_{s_k,k}[a^1]}{\overline{\pi}^1_k[a^1]} \,\bigg|\, \delta_k \right\}, \tag{53}$$

for all $a^1 \in A^1_s$, where $\delta_k := \{\hat{q}^i_{s,k}, \hat{v}^i_{s,k}\}_{(i,s)\in\{1,2\}\times S}$ includes the iterates at stage $k$, and $\omega^2_{s_k,k}[a^2]$ for $a^2 \in A^2_s$ is defined accordingly. The expectation is explicitly given by

$$\mathbb{E}\left\{ \mathbf{1}_{\{a^1_k=a^1\}} \frac{r^1_s(a^1, a^2_k) + \gamma \hat{v}^1_{s_{k+1},k} - \hat{q}^1_{s_k,k}[a^1]}{\overline{\pi}^1_k[a^1]} \,\bigg|\, \delta_k \right\}$$
$$= \overline{\pi}^1_k[a^1] \sum_{\tilde{a}^2} \overline{\pi}^2_k[\tilde{a}^2] \frac{\hat{Q}^1_{s_k,k}[a^1, \tilde{a}^2] - \hat{q}^1_{s_k,k}[a^1]}{\overline{\pi}^1_k[a^1]}, \tag{54}$$

where the auxiliary global $Q$-function estimate is as described in (47). Since the denominator disappears in (54), the stochastic approximation error is also given by

$$\omega^1_{s_k,k}[a^1] = \mathbf{1}_{\{a^1_k=a^1\}} \frac{r^1_s(a^1, a^2_k) + \gamma \hat{v}^1_{s_{k+1},k} - \hat{q}^1_{s_k,k}[a^1]}{\overline{\pi}^1_k[a^1]}$$
$$- \left( \sum_{\tilde{a}^2} \hat{Q}^1_{s_k,k}[a^1, \tilde{a}^2] \overline{\pi}^2_k[\tilde{a}^2] - \hat{q}^1_{s_k,k}[a^1] \right). \tag{55}$$

By Proposition 2, we can write (8) as

$$\hat{q}^i_{s_k,k+1} = \hat{q}^i_{s_k,k} + \alpha_{\#s_k} \left( \hat{Q}^i_{s_k,k} \overline{\pi}^{-i}_k - \hat{q}^i_{s_k,k} + \omega^i_{s_k,k} \right), \tag{56}$$

for $i = 1, 2$ if $\#s_k \geq C_{s_k}$ and Proposition 3 yields that there exists $\kappa_s \in \mathbb{N}$ such that $\#_{k'}s_k \geq C_{s_k}$ for all $k' \geq \kappa_s$.

Our goal is to characterize the limit set of this discrete-time update for every state. Like Leslie and Collins [2005], we can resort to stochastic approximation methods to transform the problem into a tractable continuous-time flow. Distinct to Markov games, we cannot characterize the convergence properties of (56) for each state separately. By (47), the update (56) yields that the current state's local Q-function estimate is coupled with any other state's value function estimate. For example, fix an arbitrary state $s$ and take a closer look at how the iterates change in-between two consecutive visits to $s$, denoted by $k$ and $k^\dagger$. Since the game does not visit state $s$ until $k^\dagger$, we have $\hat{q}^i_{s,k^\dagger} = \hat{q}^i_{s,k+1}$ and $\hat{v}^i_{s,k^\dagger} = \hat{v}^i_{s,k+1}$ for $i = 1, 2$, and $\#_{k^\dagger}s = \#_k s + 1$. In contrast, other states' value function estimates can change depending on the visits to other states at stages within the interval $(k, k^\dagger)$. Correspondingly, the iterates and the temperature parameter at $k^\dagger$ can be written in terms of the iterates at $k$ in the following compact form:

$$
\begin{bmatrix}
\hat{q}^1_{s,k^\dagger} \\
\hat{q}^2_{s,k^\dagger} \\
\hat{v}^1_{s,k^\dagger} \\
\hat{v}^2_{s,k^\dagger} \\
\vdots \\
\hat{v}^1_{s',k^\dagger} \\
\hat{v}^2_{s',k^\dagger} \\
\tau_{\#s+1}
\end{bmatrix}
=
\begin{bmatrix}
\hat{q}^1_{s,k} \\
\hat{q}^2_{s,k} \\
\hat{v}^1_{s,k} \\
\hat{v}^2_{s,k} \\
\vdots \\
\hat{v}^1_{s',k} \\
\hat{v}^2_{s',k} \\
\tau_{\#s}
\end{bmatrix}
+ \alpha_{\#s}
\left(
\begin{bmatrix}
\hat{Q}^1_{s,k}\overline{\mathrm{Br}}^2_s(\hat{q}^2_{s,k}, \tau_{\#s}) - \hat{q}^1_{s,k} \\
\hat{Q}^2_{s,k}\overline{\mathrm{Br}}^1_s(\hat{q}^1_{s,k}, \tau_{\#s}) - \hat{q}^2_{s,k} \\
0 \\
0 \\
\vdots \\
0 \\
0 \\
0
\end{bmatrix}
+
\begin{bmatrix}
\mathbf{0} \\
\mathbf{0} \\
\varepsilon^1_{s,k^\dagger} \\
\varepsilon^2_{s,k^\dagger} \\
\vdots \\
\varepsilon^1_{s',k^\dagger} \\
\varepsilon^2_{s',k^\dagger} \\
\frac{\tau_{\#s+1}-\tau_{\#s}}{\alpha_{\#s}}
\end{bmatrix}
+
\begin{bmatrix}
\omega^1_{s,k} \\
\omega^2_{s,k} \\
0 \\
0 \\
\vdots \\
0 \\
0 \\
0
\end{bmatrix}
\right), \quad (57)
$$

for all $k \geq \kappa_s$, where we define the error terms by

$$
\varepsilon^i_{s',k^\dagger} := \frac{\hat{v}^i_{s',k^\dagger} - \hat{v}^i_{s',k}}{\alpha_{\#s}}, \quad \forall(i, s') \in \{1, 2\} \times S. \tag{58}
$$

Based on Proposition 3, we can focus on asymptotic convergence properties of (57). Therefore, we are interested in when the convergence properties of (57) can be characterized through the following ordinary differential equation (in which the dynamics for $s$ is decoupled from the dynamics for any other state)

$$
\frac{dq^1_s(t)}{dt} = \bar{Q}^1_s\overline{\mathrm{Br}}^2_s(q^2_s(t), \bar{\tau}) - q^1_s(t), \tag{59a}
$$

$$
\frac{dq^2_s(t)}{dt} = \bar{Q}^2_s\overline{\mathrm{Br}}^1_s(q^1_s(t), \bar{\tau}) - q^2_s(t), \tag{59b}
$$

for some $\bar{Q}^i_s \in [-D, D]^{|A^i_s| \times |A^{-i}_s|}$ for $i \in \{1, 2\}$ and $\bar{\tau} > 0$. To this end, we can resort to Theorem 2 by showing that the limiting ordinary differential equation of (59) is given by

$$
\frac{dq^1_s(t)}{dt} = Q^1_s(t)\overline{\mathrm{Br}}^2_s(q^2_s(t), \tau_s(t)) - q^1_s(t), \tag{60a}
$$

$$
\frac{dq^2_s(t)}{dt} = Q^2_s(t)\overline{\mathrm{Br}}^1_s(q^1_s(t), \tau_s(t)) - q^2_s(t), \tag{60b}
$$

$$
\frac{dv^i_{s'}(t)}{dt} = 0, \quad \forall(i, s') \in \{1, 2\} \times S, \tag{60c}
$$

$$
\frac{d\tau_s(t)}{dt} = 0, \tag{60d}
$$

where $Q^i_s[a^i, a^{-i}](t) = r^i_s(a^1, a^2) + \gamma \sum_{s' \in S} p(s'|s, a^1, a^2)v^i_{s'}(t)$ for $i = 1, 2$. Conditions $i$ (and $ii$) in Theorem 2 are satisfied by Assumption 1-$i$ (and Proposition 1). Furthermore, the corresponding vector field is Lipschitz continous since it is continously differentiable by (7) and defined over a compact set by Proposition 1. The following two lemmas show that the conditions $iv$-$v$ listed in Theorem 2 are also satisfied. The proofs of these technical lemmas are provided in Subsection §E.4.

**Lemma 1.** *Suppose Assumption 1 and either Assumption 2 or 2' hold. Then, the stochastic approximation terms $(\omega^1_{s,k}, \omega^2_{s,k})$ satisfy (39) for all $T > 0$, $s \in S$, and $i = 1, 2$.*

Assumption 2 ensures that the denominator in the update of the local $Q$-function estimate is bounded from below by some non-zero term. On the other hand, Assumption 2' restrains the rate at which the denominator gets close to zero while letting $\lim_{c\to\infty} \tau_c = 0$.

**Lemma 2.** *Suppose Assumption 1 and either Assumption 2 or 2' hold. Then, the error terms in* (57), $\{\varepsilon_{s,k\dagger}^i\}_{(i,s)\in\{1,2\}\times S}$ *and* $(\tau_{\#s+1} - \tau_{\#s})/\alpha_{\#s}$*, are asymptotically negligible with probability* 1.

In the following step, we will zoom into (59) and formulate a Lyapunov function to characterize the limit set of not only (59) but also the original discrete-time update (57).[9]

### E.2 Zooming into local dynamics at the fast timescale

In this subsection, we focus only on (59) for an arbitrary state $s$, therefore, we drop the subscript $s$ for notational simplicity. For $t \in [0, \infty)$, we focus on the following dynamics

$$\frac{dq^1(t)}{dt} = Q^1 \overline{\mathrm{Br}}^2(q^2(t), \tau) - q^1(t), \tag{61a}$$

$$\frac{dq^2(t)}{dt} = Q^2 \overline{\mathrm{Br}}^1(q^1(t), \tau) - q^2(t), \tag{61b}$$

with arbitrary initialization of $q^i(0)$ such that $\|q^i(0)\|_\infty \leq D$, arbitrary matrices $Q^i$ such that $\|Q^i\|_{\max} \leq D$, and $\tau > 0$. The flow (61) resembles to the local $Q$-functions' evolution in the perturbed best response dynamics:

$$\frac{d\pi^1(t)}{dt} = \overline{\mathrm{Br}}^1(Q^1\pi^2(t), \tau) - \pi^1(t), \tag{62a}$$

$$\frac{d\pi^2(t)}{dt} = \overline{\mathrm{Br}}^2(Q^2\pi^1(t), \tau) - \pi^2(t), \tag{62b}$$

where $\pi^i : [0, \infty) \to \Delta(A^i)$. Indeed, they lead to the same trajectory for $(q^1, q^2)$ if we have $q^1(t) = Q^1\pi^2(t)$ and $q^2(t) = Q^2\pi^1(t)$. However, there may not always exist a strategy, e.g., $\pi^2 \in \Delta(A^2)$, such that $q^1(t) = Q^1\pi^2$. If there exists such a strategy, we say $q^1(t)$ is *belief-based*, and vice versa.

We will examine the flow (61) at a higher-dimensional space to mitigate this issue through

$$\frac{dq^1(t)}{dt} = Q^1 \overline{\mathrm{Br}}^2(q^2(t), \tau) - q^1(t), \tag{63a}$$

$$\frac{dq^2(t)}{dt} = Q^2 \overline{\mathrm{Br}}^1(q^1(t), \tau) - q^2(t), \tag{63b}$$

$$\frac{d\pi^1(t)}{dt} = \overline{\mathrm{Br}}^1(q^1(t), \tau) - \pi^1(t), \tag{63c}$$

$$\frac{d\pi^2(t)}{dt} = \overline{\mathrm{Br}}^2(q^2(t), \tau) - \pi^2(t), \tag{63d}$$

where $\pi^i(0) \in \Delta(A^i)$, for $i = 1, 2$, are initialized arbitrarily. We highlight the differences among (61), (62), and (63). In (63), the dependence between $(q^1, q^2)$ and $(\pi^1, \pi^2)$ is one direction, i.e., the evolution of $(q^1, q^2)$ is as in (61) and does not depend on $(\pi^1, \pi^2)$. On the contrary, $(\pi^1, \pi^2)$ is not some isolated process as in (62). Its evolution depends on $(q^1, q^2)$ due to $(\overline{\mathrm{Br}}^1(q^1(t), \tau), \overline{\mathrm{Br}}^2(q^2(t), \tau))$ instead of $(\overline{\mathrm{Br}}^1(Q^1\pi^2(t), \tau), \overline{\mathrm{Br}}^2(Q^2\pi^1(t), \tau))$.

We present the following continous and non-negative function as a candidate Lyapunov function for (63):

$$V(q^1, q^2, \pi^1, \pi^2) := \left[ \sum_{i=1,2} \max_{\mu \in \Delta(A^i)} \{\mu \cdot q^i + \tau\nu^i(\mu)\} - \lambda\zeta \right]_+ + \sum_{i=1,2} \|q^i - Q^i\pi^{-i}\|^2, \tag{64}$$

---

[9]Lyapunov function plays an important role to deduce convergence properties of the discrete-time update via the limiting o.d.e. because the convergence of the limiting o.d.e. does not necessarily imply the convergence of the discrete-time update in general (e.g., see Benaim [1999] and Borkar [2008]).

where we define $[g(t)]_+ := \max\{g(t), 0\}$ for a given function $g(\cdot)$, the auxiliary parameter $\lambda \in (1, 1/\gamma)$ is arbitrary, and we define

$$\zeta := \|Q^1 + (Q^2)^T\|_{\max} + \tau \log(|A^1||A^2|). \tag{65}$$

Note that $\zeta \geq 0$ depends on $Q^1, Q^2$, and $\tau$ implicitly, and it is small when the auxiliary game is close to zero-sum and the temperature parameter is close to zero. The arbitrary parameter $\lambda > 1$ plays an important role in ensuring that the set $\{(q^1, q^2, \pi^1, \pi^2) : V(q^1, q^2, \pi^1, \pi^2) = 0\}$ is a global attractor for the flow (63). Furthermore, the condition $\lambda\gamma \in (0, 1)$ will play an important role when we zoom out to the global dynamics in Subsection §E.3.

Before validating $V(\cdot)$ as a Lyapunov function, let us highlight its differences from other Lyapunov functions used for the best response dynamics with or without perturbation. For example, Hofbauer and Hopkins [2005] provided a Lyapunov function for the perturbed best response dynamics in zero-sum games and showed that such dynamics converge to a *Nash distribution* for any smooth function and any positive temperature parameter. However, we must consider arbitrary $Q^1$ and $Q^2$, which implies that $Q^1 + (Q^2)^T$ may not be a *zero matrix* in general. In other words, the underlying game is not necessarily zero-sum. Therefore, we need to consider this deviation in our candidate function.

On the other hand, Sayin et al. [2020] provided a Lyapunov function for the best response dynamics in games beyond zero-sum and showed that such dynamics converge to a bounded set with diameter depending on its deviation from a zero-sum game. Therefore, our candidate (64) has a similar flavor with the one in Sayin et al. [2020] while addressing also the perturbation and the issue induced by not being belief-based. For example, $V(q_*^1, q_*^2, \pi_*^1, \pi_*^2) = 0$ implies that

$$\sum_{i=1,2} \max_{\mu \in \Delta(A^i)} \{\mu \cdot q_*^i + \tau\nu^i(\mu)\} \leq \lambda\zeta, \tag{66}$$

which yields that

$$\sum_{i=1,2} q_*^i \cdot \overline{\mathrm{Br}}^i(q_*^i, \tau) \leq \lambda\zeta, \tag{67}$$

since the smooth functions $\nu^i(\cdot)$ are non-negative. Recall that the players update their value function estimates, e.g., $\hat{v}_{s,k}^i$, towards $\hat{q}_{s,k}^i \cdot \overline{\mathrm{Br}}_s^i(\hat{q}_{s,k}^i, \tau_{\#s})$. Therefore, such an upper bound plays an important role in chacterizing the convergence properties of the sum $\hat{v}_{s,k}^1 + \hat{v}_{s,k}^2$ and addressing the deviation from zero-sum settings. Furthermore, $V(q_*^1, q_*^2, \pi_*^1, \pi_*^2) = 0$ also yields that $q_*^i = Q^i\pi_*^{-i}$, i.e., $(q_*^1, q_*^2)$ are belief-based. It is also instructive to note that $(q_*^1, q_*^2, \pi_*^1, \pi_*^2)$ is not necessarily an equilibrium point of the flow (63). Indeed, such a Lyapunov function does not exist because the flow (63) (and best response dynamics) is not globally asymptotically stable for arbitrary matrices $(Q^1, Q^2)$.

The following lemma shows that the non-negative $V(\cdot)$ is a Lyapunov function for the flow (63) and its proof is provided in Subsection §E.4.

**Lemma 3.** *Consider any trajecttory of* (63) *and let* $x(t) := (q^1(t), q^2(t), \pi^1(t), \pi^2(t))$. *Then the candidate function* $V(\cdot)$, *as described in* (64), *satisfies*

- $V(x(t')) < V(x(t))$ *for all* $t' > t$ *if* $V(x(t)) > 0$,
- $V(x(t')) = 0$ *for all* $t' > t$ *if* $V(x(t)) = 0$.

Based on Lemma 3, we can characterize the convergence properties of the discrete-time update (57). There is a sequence of beliefs $\{\hat{\pi}_{s,k}^j \in \Delta(A_s^j)\}_{k \geq 0}$ for the sequence $\{\hat{q}_{s,k}^i\}_{k \geq 0}$ and it evolves according to

$$\hat{\pi}_{s,k+1}^j = \begin{cases} \hat{\pi}_{s,k}^j + \alpha_{\#s}\left(\overline{\pi}_k^j - \hat{\pi}_{s,k}^j\right) & \text{if } s = s_k \\ \hat{\pi}_{s,k}^j & \text{o.w.} \end{cases} \tag{68}$$

with some arbitrary initialization, and satisfies

$$\lim_{k \to \infty} \|\hat{q}_{s,k}^i - \hat{Q}_{s,k}^i \hat{\pi}_{s,k}^j\|^2 = 0, \tag{69}$$

where $j \neq i$. Denote $\bar{Q}_{s,k} := \bar{Q}^1_{s,k} + (\bar{Q}^2_{s,k})^T$. Then, Lemma 3 yields that

$$\lim_{k \to \infty} \left[ \sum_{i=1,2} \left( \hat{q}^i_{s,k} \cdot \bar{\pi}^i_k + \tau_{\#s} \nu^i_s(\bar{\pi}^i_k) \right) - \lambda \left( \|\bar{Q}_{s,k}\|_{\max} + \tau_{\#s} \log(|A^1_s||A^2_s|) \right) \right]_+ = 0,$$

which implies that there exists $\{\bar{e}_{s,k} \geq 0\}_{k \geq 0}$ and $\lim_{k \to \infty} \bar{e}_{s,k} = 0$ such that

$$\sum_{i=1,2} \left( \hat{q}^i_{s,k} \cdot \bar{\pi}^i_k + \tau_{\#s} \nu^i_s(\bar{\pi}^i_k) \right) \leq \lambda \left( \|\bar{Q}_{s,k}\|_{\max} + \tau_{\#s} \log(|A^1_s||A^2_s|) \right) + \bar{e}_{s,k}, \quad \forall k \geq 0. \quad (70)$$

In the following, we characterize the convergence properties of the value function estimates based on (69) and (70), respectively, showing that the local $Q$-function estimates are asymptotically belief-based and characterizing an upper bound on the sum of the (perturbed) values.

### E.3  Zooming out to global dynamics at the slow timescale

Next, we focus on the evolution of the value function estimates. To this end, we first consider how the sum of the players' value function estimates specific to state $s$ (denoted by $\bar{v}_{s,k} := \hat{v}^1_{s,k} + \hat{v}^2_{s,k}$) evolves:

$$\bar{v}_{s,k+1} = \begin{cases} \bar{v}_{s,k} + \beta_{\#s} \left[ \sum_{i=1,2} \bar{\pi}^i_k \cdot \hat{q}^i_{s,k} - \bar{v}_{s,k} \right] & \text{if } s = s_k \\ \bar{v}_{s,k} & \text{o.w.} \end{cases} \quad (71)$$

We can view (71) as the sum $\bar{v}_{s,k}$ moving toward (or tracking) the target $\sum_{i=1,2} \bar{\pi}^i_k \cdot \hat{q}^i_{s,k}$. The target is bounded from above by

$$\sum_{i=1,2} \bar{\pi}^i_k \cdot \hat{q}^i_{s,k} \leq \lambda \|\bar{Q}_{s,k}\|_{\max} + \lambda \tau_{\#s} \log(|A^1_s||A^2_s|) + \bar{e}_{s,k}, \quad \forall k \geq 0, \quad (72)$$

by (70). We can also bound the target from below by using the smooth best response definition and (69) as follows:

$$\sum_{i=1,2} \left( \bar{\pi}^i_k \cdot \hat{q}^i_{s,k} + \tau_{\#s} \nu^i_s(\bar{\pi}^i_{s,k}) \right) \geq (\hat{\pi}^1_{s,k})^T \bar{Q}_{s,k} \hat{\pi}^2_{s,k} + \tau_{\#s} \sum_{i=1,2} \nu^i_s(\hat{\pi}^i_{s,k}) + \underline{e}_{s,k}, \quad \forall k \geq 0, \quad (73)$$

where the error term is given by

$$\underline{e}_{s,k} := \sum_{i=1,2} (\hat{\pi}^i_{s,k})^T (\hat{q}^i_{s,k} - \hat{Q}^i_{s,k} \hat{\pi}^{-i}_{s,k}) \quad (74)$$

and it is asymptotically negligible by (69). Since $\hat{\pi}^i_{s,k} \in \Delta(A^i_s)$, we obtain

$$\sum_{i=1,2} \bar{\pi}^i_k \cdot \hat{q}^i_{s,k} \geq -\|\bar{Q}_{s,k}\|_{\max} + \tau_{\#s} \sum_{i=1,2} \left( \nu^i_s(\hat{\pi}^i_{s,k}) - \nu^i_s(\bar{\pi}^i_k) \right) + \underline{e}_{s,k} \quad (75)$$

$$\geq -\lambda \|\bar{Q}_{s,k}\|_{\max} - \lambda \tau_{\#s} \log(|A^1_s||A^2_s|) + \underline{e}_{s,k}, \quad (76)$$

where the last inequality follows since $\lambda > 1$ and $\nu^i_s : \Delta(A^i_s) \to [0, \log(|A^i_s|)]$.

Based on the fact that $r^1_s(a^1, a^2) + r^2_s(a^1, a^2) = 0$ for all $(a^1, a^2)$, we can formulate a bound on $\|\bar{Q}_{s,k}\|_{\max}$ from above in terms of $\{\bar{v}_{s',k}\}_{s' \in S}$ as follows:

$$\|\bar{Q}_{s,k}\|_{\max} = \max_{(a^1,a^2)} \left| r^1_s(a^1, a^2) + r^2_s(a^1, a^2) + \gamma \sum_{s' \in S} p(s'|s, a^1, a^2) \bar{v}_{s',k} \right|$$

$$\leq \gamma \max_{s' \in S} |\bar{v}_{s',k}|. \quad (77)$$

Combining (72), (76), and (77), we obtain

$$-\lambda \gamma \max_{s' \in S} |\bar{v}_{s',k}| - \lambda \tau_{\#s} \log(|A^1_s||A^2_s|) + \underline{e}_{s,k} \leq \sum_{i=1,2} \bar{\pi}^i_k \cdot \hat{q}^i_{s,k}$$

$$\leq \lambda \gamma \max_{s' \in S} |\bar{v}_{s',k}| + \lambda \tau_{\#s} \log(|A^1_s||A^2_s|) + \bar{e}_{s,k}, \quad (78)$$

for all $s \in S$ and $k \geq 0$, with some asymptotically negligible error terms $\underline{e}_{s,k}$ and $\overline{e}_{s,k}$. The condition $\lambda\gamma \in (0,1)$ yields that the target in (71) shrinks in absolute value as $k \to \infty$. Based on (71) and Theorem 3, we obtain

$$\limsup_{k\to\infty} \max_{s\in S} |\bar{v}_{s,k}| \leq \frac{1}{1-\lambda\gamma}\lambda\xi \lim_{c\to\infty} \tau_c, \tag{79}$$

where $\xi := \max_{s\in S}\{\log(|A_s^1||A_s^2|)\}$ since $\#s \to \infty$ as $k \to \infty$ with probability 1 by Proposition 3. Therefore, the auxiliary games get close to (or become) zero-sum asymptotically like the two-timescale fictitious play in Sayin et al. [2020] but with a radically uncoupled scheme.

The next and last step is about characterizing the asymptotic behavior of the tracking error (45):

$$\limsup_{k\to\infty} \left|\bar{\pi}_{s,k}^1 \cdot \hat{q}_{s,k}^1 - \text{val}^1(\hat{Q}_{s,k}^1)\right|$$

$$\overset{(a)}{\leq} \limsup_{k\to\infty} \left|\max_{\mu^1}\left\{(\mu^1)^T\hat{Q}_{s,k}^1\hat{\pi}_{s,k}^2\right\} - \text{val}^1(\hat{Q}_{s,k}^1)\right|$$

$$+ \limsup_{k\to\infty} \left|\max_{\mu^1}\left\{(\mu^1)^T\hat{Q}_{s,k}^1\hat{\pi}_{s,k}^2\right\} - \bar{\pi}_{s,k}^1 \cdot \hat{q}_{s,k}^1\right|$$

$$\overset{(b)}{\leq} \limsup_{k\to\infty} \left|\max_{\mu^1}\left\{(\mu^1)^T\hat{Q}_{s,k}^1\hat{\pi}_{s,k}^2\right\} - \text{val}^1(\hat{Q}_{s,k}^1)\right| + \xi \lim_{c\to\infty} \tau_c$$

$$\overset{(c)}{\leq} \limsup_{k\to\infty} \left|\max_{\mu^1}\left\{(\mu^1)^T\hat{Q}_{s,k}^1\hat{\pi}_{s,k}^2\right\} - \min_{\mu^2}\left\{(\hat{\pi}_{s,k}^1)^T\hat{Q}_{s,k}^1\mu^2\right\}\right| + \xi \lim_{c\to\infty} \tau_c$$

$$\overset{(d)}{\leq} \limsup_{k\to\infty} \left|\max_{\mu^2}\left\{(\mu^2)^T\hat{Q}_{s,k}^2\hat{\pi}_{s,k}^1\right\} + \min_{\mu^2}\left\{(\mu^2)^T(\hat{Q}_{s,k}^1)^T\hat{\pi}_{s,k}^1\right\}\right|$$

$$+ \limsup_{k\to\infty} \left|\sum_{i=1,2} \max_{\mu^i}\{\mu^i \cdot \hat{Q}_{s,k}^i\hat{\pi}_{s,k}^{-i}\}\right| + \xi \lim_{c\to\infty} \tau_c$$

$$\overset{(e)}{\leq} \limsup_{k\to\infty} \|\bar{Q}_{s,k}\|_{\max} + \limsup_{k\to\infty} \left|\sum_{i=1,2} \max_{\mu^i}\{\mu^i \cdot \hat{Q}_{s,k}^i\hat{\pi}_{s,k}^{-i}\}\right| + \xi \lim_{c\to\infty} \tau_c$$

$$\overset{(f)}{\leq} \gamma\limsup_{k\to\infty} \max_{s'\in S} |\bar{v}_{s',k}| + \limsup_{k\to\infty} \left|\sum_{i=1,2} \max_{\mu^i}\{\mu^i \cdot \hat{Q}_{s,k}^i\hat{\pi}_{s,k}^{-i}\}\right| + \xi \lim_{c\to\infty} \tau_c$$

$$\overset{(g)}{\leq} \gamma\limsup_{k\to\infty} \max_{s'\in S} |\bar{v}_{s',k}| + \limsup_{k\to\infty} \left|\sum_{i=1,2} \max_{\mu^i}\{\mu^i \cdot \hat{q}_{s,k}^i\}\right| + \xi \lim_{c\to\infty} \tau_c$$

$$\overset{(h)}{\leq} \gamma\limsup_{k\to\infty} \max_{s'\in S} |\bar{v}_{s',k}| + \limsup_{k\to\infty} \left|\sum_{i=1,2} \bar{\pi}_k^i \cdot \hat{q}_{s,k}^i\right| + 2\xi \lim_{c\to\infty} \tau_c$$

$$\overset{(i)}{\leq} (\lambda\gamma + \gamma)\limsup_{k\to\infty} \max_{s\in S} |\bar{v}_{s,k}| + (\lambda + 2)\xi \lim_{c\to\infty} \tau_c$$

$$\overset{(j)}{\leq} \left(\frac{\lambda(\lambda\gamma + \gamma)}{1-\lambda\gamma} + (\lambda + 2)\right)\xi \lim_{c\to\infty} \tau_c$$

$$= \left(\frac{2 + \lambda - \lambda\gamma}{1-\lambda\gamma}\right)\xi \lim_{c\to\infty} \tau_c. \tag{80}$$

Particularly, $(a)$ follows from triangle inequality; $(b)$ follows since

$$\bar{\pi}_k^i \cdot \hat{q}_{s,k}^i + \tau_{\#s} \log(|A_s^i|) \geq \max_{\mu^i\in\Delta(A_s^i)}\{\mu^i \cdot \hat{q}_{s,k}^i\} \geq \bar{\pi}_k^i \cdot \hat{q}_{s,k}^i, \tag{81}$$

by definition of best response and smooth best response and since $\hat{q}_{s,k}^i$ is asymptotically belief-based and $\max\{\cdot\}$ is a continuous operator; $(c)$ follows from the fact that

$$\max_{\mu^1}\left\{(\mu^1)^T\hat{Q}_{s,k}^1\hat{\pi}_{s,k}^2\right\} \geq \text{val}^1(\hat{Q}_{s,k}^1) \geq \min_{\mu^2}\left\{(\hat{\pi}_{s,k}^1)^T\hat{Q}_{s,k}^1\mu^2\right\}; \tag{82}$$

$(d)$ follows from the triangle inequality; $(e)$ follows since we have

$$\left| \max_{\mu^2} \left\{ (\mu^2)^T \hat{Q}^2_{s,k} \hat{\pi}^1_{s,k} \right\} + \min_{\mu^2} \left\{ (\mu^2)^T (\hat{Q}^1_{s,k})^T \hat{\pi}^1_{s,k} \right\} \right|$$

$$= \left| \max_{\mu^2} \left\{ (\mu^2)^T \hat{Q}^2_{s,k} \hat{\pi}^1_{s,k} \right\} - \max_{\mu^2} \left\{ (\mu^2)^T (-\hat{Q}^1_{s,k})^T \hat{\pi}^1_{s,k} \right\} \right| \tag{83}$$

$$\leq \|\bar{Q}_{s,k}\|_{\max}, \tag{84}$$

where (83) corresponds to the difference between the maximum values player 2 would get in the scenarios in which it has the payoff matrices $\hat{Q}^2_{s,k}$ versus $(-\hat{Q}^1_{s,k})$ in an auxiliary strategic-form game, given that the opponent's play is fixed, and this difference is bounded from above by $\|\hat{Q}^2_{s,k} - (-\hat{Q}^1_{s,k})\|_{\max} = \|\bar{Q}_{s,k}\|_{\max}$; $(f)$ follows from (77); $(g)$ follows since $\hat{q}^i_{s,k}$ is asymptotically belief-based; $(h)$ follows from (81); $(i)$ follows since (78) yields that

$$\limsup_{k\to\infty} \left| \sum_{i=1,2} \bar{\pi}^i_k \cdot \hat{q}^i_{s,k} \right| \leq \lambda\gamma \limsup_{k\to\infty} \max_{s\in S} |\bar{v}_{s,k}| + \lambda\xi \lim_{c\to\infty} \tau_c; \tag{85}$$

and finally $(j)$ follows from (79). This completes the first part of the result on the asymptotic behavior of $\{\hat{v}^i_{s,k}\}_{k\geq 0}$.

On the other hand, the weighted time-average of smoothed best responses corresponds to $\hat{\pi}^i_{s,k}$, as described iteratively in (68). Note that $v^i_{\pi_*}(s) = \mathrm{val}^i(Q^i_{\pi_*}(s,\cdot))$ by its definition. Therefore, for any $s$, we define $v^{-i}_{*,\pi^{-i}}(s) := \min_{\pi^i} v^{-i}_{\pi^i,\pi^{-i}}(s) = -\max_{\pi^i} v^i_{\pi^i,\pi^{-i}}(s) =: -v^i_{*,\pi^{-i}}(s)$. Note that the convention is that for agent $i$ and her value function $v^i_{\pi^i,\pi^{-i}}$, we use $v^i_{*,\pi^{-i}}$ to denote the $\max$ over her own strategy $\pi^i$, and $v^i_{\pi^i,*}$ to denote the $\min$ over her opponent's strategy $\pi^{-i}$. In other words, agent $i$ always maximizes her value, while her opponent always minimizes it. Also note that for fixed strategy of one player, the problem is a Markov decision process, which always admits some maximizing/minimizing strategy for all $s$, i.e., these best-response values are well-defined. Finally, for these value functions, one can define the corresponding $Q$-functions satisfying the following Bellman equations:

$$Q^i_{*,\pi^{-i}}(s,a^1,a^2) = r^i_s(a^1,a^2) + \gamma \sum_{s'\in S} v^i_{*,\pi^{-i}}(s') p(s'|s,a^1,a^2), \quad \forall(s,a^1,a^2)$$

$$v^i_{*,\pi^{-i}}(s) = \max_{\mu} \mathbb{E}_{a^i\sim\mu,\, a^{-i}\sim\pi^{-i}_s} \left[ Q^i_{*,\pi^{-i}}(s,a^1,a^2) \right], \quad \forall\, s,$$

and

$$Q^i_{\pi_*}(s,a^1,a^2) = r^i_s(a^1,a^2) + \gamma \sum_{s'\in S} v^i_{\pi_*}(s') p(s'|s,a^1,a^2), \quad \forall(s,a^1,a^2)$$

$$v^i_{\pi_*}(s) = \max_{\mu} \min_{\nu} \mathbb{E}_{a^i\sim\mu,\, a^{-i}\sim\nu} \left[ Q^i_{\pi_*}(s,a^1,a^2) \right], \quad \forall\, s.$$

Other quantities can be defined similarly.

Therefore, we have

$$0 \leq v_{\pi_*}^{-i}(s) - v_{*,\hat{\pi}_k^{-i}}^{-i}(s) = -\max_{\mu^i}(\mu^i)^T Q_{\pi_*}^i(s,\cdot)\pi_{s,*}^{-i} + \max_{\mu^i}(\mu^i)^T Q_{*,\hat{\pi}_k^{-i}}^i(s,\cdot)\hat{\pi}_{s,k}^{-i} \tag{86}$$

$$\leq \left| \max_{\mu^i}(\mu^i)^T Q_{*,\hat{\pi}_k^{-i}}^i(s,\cdot)\hat{\pi}_{s,k}^{-i} - \max_{\mu^i}(\mu^i)^T Q_{\pi_*}^i(s,\cdot)\hat{\pi}_{s,k}^{-i} \right|$$
$$+ \left| \max_{\mu^i}(\mu^i)^T Q_{\pi_*}^i(s,\cdot)\hat{\pi}_{s,k}^{-i} - \max_{\mu^i}(\mu^i)^T Q_{\pi_*}^i(s,\cdot)\pi_{s,*}^{-i} \right| \tag{87}$$

$$\leq \gamma \cdot \max_{s \in S} \left| v_{*,\hat{\pi}_k^{-i}}^i(s) - v_{\pi_*}^i(s) \right| + \left| \mathrm{val}^i(\hat{Q}_{s,k}^i) - \max_{\mu^i}(\mu^i)^T Q_{\pi_*}^i(s,\cdot)\pi_{s,*}^{-i} \right|$$
$$+ \left| \max_{\mu^i}(\mu^i)^T Q_{\pi_*}^i(s,\cdot)\hat{\pi}_{s,k}^{-i} - \mathrm{val}^i(\hat{Q}_{s,k}^i) \right| \tag{88}$$

$$\leq \gamma \cdot \max_{s \in S} \left| v_{\pi_*}^{-i}(s) - v_{*,\hat{\pi}_k^{-i}}^{-i}(s) \right| + \left| \mathrm{val}^i(\hat{Q}_{s,k}^i) - \max_{\mu^i}(\mu^i)^T Q_{\pi_*}^i(s,\cdot)\pi_{s,*}^{-i} \right|$$
$$+ \left| \max_{\mu^i}(\mu^i)^T Q_{\pi_*}^i(s,\cdot)\hat{\pi}_{s,k}^{-i} - \max_{\mu^i}(\mu^i)^T \hat{Q}_{s,k}^i(s,\cdot)\hat{\pi}_{s,k}^{-i} \right|$$
$$+ \left| \max_{\mu^i}(\mu^i)^T \hat{Q}_{s,k}^i(s,\cdot)\hat{\pi}_{s,k}^{-i} - \mathrm{val}^i(\hat{Q}_{s,k}^i) \right|, \tag{89}$$

where (86) is due to one-step Bellman equation and the zero-sum structure of the underlying game, (87) follows by inserting $\max_{\mu^i}(\mu^i)^T Q_{\pi_*}^i(s,\cdot)\hat{\pi}_{s,k}^{-i}$, (88) follows by the fact that

$$\left| \max_{\mu^i}(\mu^i)^T Q_{*,\hat{\pi}_k^{-i}}^i(s,\cdot)\hat{\pi}_{s,k}^{-i} - \max_{\mu^i}(\mu^i)^T Q_{\pi_*}^i(s,\cdot)\hat{\pi}_{s,k}^{-i} \right|$$
$$\leq \max_{\mu^i} \left| (\mu^i)^T Q_{*,\hat{\pi}_k^{-i}}^i(s,\cdot)\hat{\pi}_{s,k}^{-i} - (\mu^i)^T Q_{\pi_*}^i(s,\cdot)\hat{\pi}_{s,k}^{-i} \right| \leq \gamma \cdot \max_{s \in S} \left| v_{*,\hat{\pi}_k^{-i}}^i(s) - v_{\pi_*}^i(s) \right|$$

and by inserting $\mathrm{val}^i(\hat{Q}_{s,k}^i)$, and finally, (89) follows by inserting $\max_{\mu^i}(\mu^i)^T \hat{Q}_{s,k}^i(s,\cdot)\hat{\pi}_{s,k}^{-i}$, and by the zero-sum property of the values. The last three terms in (89) can be further bounded as follows:
1) by definitions, we have

$$\left| \mathrm{val}^i(\hat{Q}_{s,k}^i) - \max_{\mu^i}(\mu^i)^T Q_{\pi_*}^i(s,\cdot)\pi_{s,*}^{-i} \right| \leq \left\| \hat{Q}_{s,k}^i - Q_{\pi_*}^i(s,\cdot) \right\|_{\max}$$
$$\leq \gamma \max_{s' \in S} |v_{\pi_*}^i(s') - \hat{v}_{s',k}^i|; \tag{90}$$

$$\left| \max_{\mu^i}(\mu^i)^T Q_{\pi_*}^i(s,\cdot)\hat{\pi}_{s,k}^{-i} - \max_{\mu^i}(\mu^i)^T \hat{Q}_{s,k}^i(s,\cdot)\hat{\pi}_{s,k}^{-i} \right|$$
$$\leq \left\| \hat{Q}_{s,k}^i - Q_{\pi_*}^i(s,\cdot) \right\|_{\max} \leq \gamma \max_{s' \in S} |v_{\pi_*}^i(s') - \hat{v}_{s',k}^i|; \tag{91}$$

2) by $(c)$-$(i)$ in (80), we have

$$\limsup_{k \to \infty} \left| \max_{\mu^i}(\mu^i)^T \hat{Q}_{s,k}^i \hat{\pi}_{s,k}^{-i} - \mathrm{val}^i(\hat{Q}_{s,k}^i) \right| \leq \left( \frac{1+\lambda}{1-\lambda\gamma} \right) \xi \lim_{c \to \infty} \tau_c. \tag{92}$$

Hence, combining (89)-(92), we have

$$\limsup_{k \to \infty} \max_{s \in S} \left| v_{\pi_*}^{-i}(s) - v_{*,\hat{\pi}_k^{-i}}^{-i}(s) \right| \leq \frac{1}{1-\gamma} \cdot \left[ 2\gamma \limsup_{k \to \infty} \max_{s' \in S} |v_{\pi_*}^i(s') - \hat{v}_{s',k}^i| + \left( \frac{1+\lambda}{1-\lambda\gamma} \right) \xi \lim_{c \to \infty} \tau_c \right]$$
$$\leq \frac{1}{1-\gamma} \cdot \left[ 2\gamma\xi g(\gamma) \lim_{c \to \infty} \tau_c + \left( \frac{1+\lambda}{1-\lambda\gamma} \right) \xi \lim_{c \to \infty} \tau_c \right]. \tag{93}$$

Finally, notice the fact that

$$0 \leq v_{*,\hat{\pi}_k^{-i}}^i(s) - v_{\hat{\pi}_k^i,\hat{\pi}_k^{-i}}^i(s) \leq v_{*,\hat{\pi}_k^{-i}}^i(s) - v_{\hat{\pi}_k^i,*}^i(s) = -v_{*,\hat{\pi}_k^{-i}}^{-i}(s) + v_{\hat{\pi}_k^i,*}^{-i}(s) \tag{94}$$
$$= -v_{\hat{\pi}_k^i,*}^i(s) + v_{\pi_*}^i(s) + v_{\pi_*}^{-i}(s) - v_{*,\hat{\pi}_k^{-i}}^{-i}(s)$$
$$\leq \max_{s \in S} \left| v_{\pi_*}^i(s) - v_{\hat{\pi}_k^i,*}^i(s) \right| + \max_{s \in S} \left| v_{\pi_*}^{-i}(s) - v_{*,\hat{\pi}_k^{-i}}^{-i}(s) \right|_{\max},$$

where the last inequality follows from (93) and its counterpart by switching the role of $-i$ and $i$ therein. Combined with the definition of $\epsilon$-Nash equilibrium with (93)-(94), we complete the proof.

### E.4 Proofs of Technical Lemmas 1-3

**Lemma 1**. *Suppose Assumption 1 and either Assumption 2 or 2' hold. Then, the stochastic approximation terms $(\omega^1_{s,k}, \omega^2_{s,k})$ satisfy (39) for all $T > 0$, $s \in S$, and $i = 1, 2$.*

*Proof:* By (53), the stochastic approximation term $\omega^1_{s,k}[a^1]$ (and $\omega^2_{s,k}$) can be written as

$$\omega^1_{s,k}[a^1] = \frac{\tilde{\omega}^1_{s,k}[a^1]}{\pi^1_k[a^1]}, \quad \forall a^1 \in A^1_s, \tag{95}$$

where we define

$$\begin{aligned}
\tilde{\omega}^1_{s,k}[a^1] &:= \mathbf{1}_{\{a^1_k=a^1\}}(r^1_s(a^1, a^2_k) + \gamma\hat{v}^1_{s_{k+1},k} - \hat{q}^1_{s,k}[a^1]) \\
&\quad - \mathbb{E}\left\{ \mathbf{1}_{\{a^1_k=a^1\}}(r^1_s(a^1, a^2_k) + \gamma\hat{v}^1_{s_{k+1},k} - \hat{q}^1_{s,k}[a^1]) \,\Big|\, \delta_k \right\} \tag{96} \\
&= \mathbf{1}_{\{a^1_k=a^1\}}(r^1_s(a^1, a^2_k) + \gamma\hat{v}^1_{s_{k+1},k}) \\
&\quad - \mathbb{E}\left\{ \mathbf{1}_{\{a^1_k=a^1\}}(r^1_s(a^1, a^2_k) + \gamma\hat{v}^1_{s_{k+1},k}) \,\Big|\, \delta_k \right\}, \tag{97}
\end{aligned}$$

which is a square-integrable Martingale difference sequence since the iterates remain bounded by Proposition 1. Then, the proof follows from the proof of [Benaim, 1999, Proposition 4.2] by substituting the step size $\alpha_{\#s}$ with $\frac{\alpha_{\#s}}{\pi^i_k[a^i]}$ and showing that

$$\sum_{k \geq 0 : s = s_k} \left( \frac{\alpha_{\#s}}{\pi^i_k[a^i]} \right)^2 < \infty, \quad \forall a^i \in A^i_s \text{ and } s \in S. \tag{98}$$

If Assumption 2 holds, then the analytical form of $\overline{\pi}^i_k[a^i]$, as described in (7), yields that $\overline{\pi}^i_k[a^i] \geq \frac{1}{|A^i_s|}\exp(-2D/\epsilon)$. Correspondingly, the sum in (98) is bounded from above by

$$\sum_{k \geq 0 : s = s_k} \left( \frac{\alpha_{\#s}}{\pi^i_k[a^i]} \right)^2 \leq |A^i_s|^2 \exp\left( \frac{4D}{\epsilon} \right) \sum_{k \geq 0 : s = s_k} \alpha^2_{\#s}. \tag{99}$$

The right-hand side is a convergent sum by Assumption 2-ii.

On the other hand, if Assumption 2' holds, then we no longer have a fixed lower bound on $\overline{\pi}^i_k[a^i]$. Instead, we have $\overline{\pi}^i_k[a^i] \geq \frac{1}{|A^i_s|}\exp\{-2D/\tau_{\#s}\}$. Correspondingly, the sum in (98) is now bounded from above by

$$\sum_{k \geq 0 : s = s_k} \left( \frac{\alpha_{\#s}}{\pi^i_k[a^i]} \right)^2 \leq |A^i_s|^2 \sum_{k \geq 0 : s = s_k} \alpha^2_{\#s} \exp\left( \frac{4D}{\tau_{\#s}} \right). \tag{100}$$

By Assumption 2'-ii, we have $\exp(4D/\tau_c) < C'\alpha_c^{-\rho}$ for all $c \geq C$. Therefore, we obtain

$$\sum_{k \geq 0 : s = s_k} \left( \frac{\alpha_{\#s}}{\pi^i_k[a^i]} \right)^2 \leq |A^i_s|^2 \sum_{c=0}^{C-1} \alpha_c^2 \exp\left( \frac{4D}{\tau_c} \right) + |A^i_s|^2 C' \sum_{c \geq C} \alpha_c^{2-\rho}. \tag{101}$$

The right-hand side is a convergent sum by Assumption 2'-ii. This completes the proof. $\square$

**Lemma 2**. *Suppose Assumption 1 and either Assumption 2 or 2' hold. Then, the error terms in (57), $\{\varepsilon^i_{s,k^\dagger}\}_{(i,s)\in\{1,2\}\times S}$ and $(\tau_{\#s+1} - \tau_{\#s})/\alpha_{\#s}$, are asymptotically negligible with probability 1.*

*Proof:* The error term $(\tau_{\#s+1} - \tau_{\#s})/\alpha_{\#s}$ is asymptotically negligible with probability 1 either by Assumption 2-ii or Assumption 2'-ii. On the other hand, the definition of $\varepsilon^i_{s',k^\dagger}$, as described in (58), and the evolution of $\hat{v}^i_{s',k}$ yield that

$$\varepsilon^i_{s',k^\dagger} = \frac{\sum_{l=k}^{k^\dagger-1} \mathbf{1}_{\{s_l=s'\}}\beta_{\#l s'}(\hat{q}^i_{s',l} \cdot \overline{\pi}^i_l - \hat{v}^i_{s',l})}{\alpha_{\#_k s}}, \quad \forall s' \in S. \tag{102}$$

Since $\{\beta_c\}_{c>0}$ is non-increasing by Assumption 1-i and the iterates are bounded by $D$ by Proposition 1, the error term $\varepsilon^i_{s',k^\dagger}$ is bounded from above by

$$|\varepsilon^i_{s',k^\dagger}| \le \frac{\beta_{\#_k s'}}{\alpha_{\#_k s}}(k^\dagger - k)2D, \quad \forall s' \in S. \tag{103}$$

By Borel-Cantelli Lemma, if we have

$$\sum_{k=0}^{\infty} \mathbb{P}\left\{\frac{\beta_{\#_k s'}}{\alpha_{\#_k s}}(k^\dagger - k) > \lambda\right\} < \infty \tag{104}$$

for any $\lambda > 0$, then $|\varepsilon^i_{s',k^\dagger}| \to 0$ as $k \to \infty$ with probability 1. To this end, we will focus on the argument of the summation (104). Since $\#_k s \le k$ and $\{\alpha_c\}_{c>0}$ is a non-increasing sequence by Assumption 1-i, we have

$$\mathbb{P}\left\{\frac{\beta_{\#_k s'}}{\alpha_{\#_k s}}(k^\dagger - k) > \lambda\right\} \le \mathbb{P}\left\{\frac{\beta_{\#_k s'}}{\alpha_k}(k^\dagger - k) > \lambda\right\}, \tag{105}$$

$$= \sum_{\kappa=1}^{\infty} \mathbb{P}\left\{k^\dagger - k = \kappa\right\} \sum_{l=0: \frac{\beta_l}{\alpha_k} > \frac{\lambda}{\kappa}}^{k} \mathbb{P}\left\{l = \#_k s' \mid k^\dagger - k = \kappa\right\}, \tag{106}$$

$$= \sum_{\kappa=1}^{\infty} \mathbb{P}\left\{k^\dagger - k = \kappa\right\} \mathbb{P}\left\{\#_k s' \le \ell_k\left(\frac{\lambda}{\kappa}\right) \mid k^\dagger - k = \kappa\right\}, \tag{107}$$

where $\ell_k\left(\frac{\lambda}{\kappa}\right) := \max\left\{l \in \Xi_k\left(\frac{\lambda}{\kappa}\right)\right\}$ with $\Xi_k(\lambda) := \max\left\{l \in \mathbb{Z} : l \le k \text{ and } \frac{\beta_l}{\alpha_k} > \lambda\right\} \cup \{-1\}$. Therefore, we obtain

$$\sum_{k=0}^{\infty} \mathbb{P}\left\{\frac{\beta_{\#_k s'}}{\alpha_{\#_k s}}(k^\dagger - k) > \lambda\right\} \le \sum_{k=0}^{\infty}\sum_{\kappa=1}^{\infty} \mathbb{P}\left\{k^\dagger - k = \kappa\right\} \mathbb{P}\left\{\#_k s' \le \ell_k\left(\frac{\lambda}{\kappa}\right) \mid k^\dagger - k = \kappa\right\}. \tag{108}$$

Since the argument of the summation on the right-hand side is non-negative, convergence of the following:

$$\sum_{\kappa=1}^{\infty}\sum_{k=0}^{\infty} \mathbb{P}\left\{\kappa = k^\dagger - k\right\} \mathbb{P}\left\{\#_k s' \le \ell_k\left(\frac{\lambda}{\kappa}\right) \mid k^\dagger - k = \kappa\right\}$$

$$= \sum_{\kappa=1}^{\infty} \mathbb{P}\left\{\kappa = k^\dagger - k\right\} \sum_{k=0}^{\infty} \mathbb{P}\left\{\#_k s' \le \ell_k\left(\frac{\lambda}{\kappa}\right) \mid k^\dagger - k = \kappa\right\}, \tag{109}$$

where the order of summations is interchanged would imply the convergence of (108). Correspondingly, we will show that (109) is convergent instead.

Partition the time axis into $n$-stage intervals and define $\bar{\#}^n_k s'$ as a counting process that increases by 1 at the end of an $n$-stage interval if $s'$ is visited at least once within the last $n$-stage. More precisely, $\bar{\#}^n_k s'$ is, recursively, given by

$$\bar{\#}^n_{k+1} s' = \begin{cases} \bar{\#}^n_k s' + 1 & \text{if } 0 \equiv k \mod n \text{ and } \exists l \in (k-n, k] : s_l = s' \\ \bar{\#}^n_k s' & \text{o.w.} \end{cases} \tag{110}$$

By its definition, we have $\bar{\#}^n_k s' \le \#_k s'$, and therefore, we obtain

$$\mathbb{P}\left\{\#_k s' \le \ell_k\left(\frac{\lambda}{\kappa}\right) \mid k^\dagger - k = \kappa\right\} \le \mathbb{P}\left\{\bar{\#}^n_k s' \le \ell_k\left(\frac{\lambda}{\kappa}\right) \mid k^\dagger - k = \kappa\right\}. \tag{111}$$

Either Assumption 2 or 2' yield that the probability that state $s'$ is visited within an $n$-stage interval is bounded from below, e.g., by some $\underline{p} > 0$, for every sequence of actions. Correspondingly, Correspondingly, the probability that $s'$ is not visited within the $n$-stage interval is bounded from above by $1 - \underline{p}$. Therefore, we can bound the right-hand side of (111) from above by

$$\mathbb{P}\left\{\bar{\#}^n_k s' \le \ell_k\left(\frac{\lambda}{\kappa}\right) \mid k^\dagger - k = \kappa\right\} \le \sum_{l=0: l \le \ell_k\left(\frac{\lambda}{\kappa}\right)}^{\lfloor \frac{k}{n} \rfloor} \binom{\lfloor \frac{k}{n} \rfloor}{l} 1^l (1-\underline{p})^{\lfloor \frac{k}{n} \rfloor - l}. \tag{112}$$

Assumption 1-ii yields that for any $M \in (0,1)$, there exists a non-decreasing polynomial function $C_M(\cdot)$ such that

$$\ell_k\left(\frac{\lambda}{\kappa}\right) \le Mk < nM\left(\left\lfloor\frac{k}{n}\right\rfloor + 1\right), \quad \forall k \ge C_M\left(\frac{\kappa}{\lambda}\right). \tag{113}$$

For $k \ge n$, (113) can also be written as

$$\frac{\ell_k(\lambda/\kappa)}{\lfloor k/n \rfloor} \le nM\left(1 + \frac{1}{\lfloor k/n \rfloor}\right) \le 2nM, \quad \forall l \ge \max\{C_M(\kappa/\lambda), n\}. \tag{114}$$

Since $M \in (0,1)$ is arbitrary, there exists $M$ such that $2nM < 1$, i.e, $M \in (0, 1/2n)$. Therefore, we have

$$\sum_{l=0:l\le\ell_k\left(\frac{\lambda}{\kappa}\right)}^{\lfloor\frac{k}{n}\rfloor} \binom{\lfloor\frac{k}{n}\rfloor}{l} 1^l(1-\underline{p})^{\lfloor\frac{k}{n}\rfloor - l} \le (1-\underline{p})^{\lfloor\frac{k}{n}\rfloor - \ell_k\left(\frac{\lambda}{\kappa}\right)} \sum_{l=0}^{\ell_k\left(\frac{\lambda}{\kappa}\right)} \binom{\lfloor\frac{k}{n}\rfloor}{l}, \tag{115}$$

for all $k \ge \max\{C_M(\kappa/\lambda), n\}$, since $(1-\underline{p}) \in [0,1)$. Furthermore, we can set $M \in (0, 1/2n)$ such that $2nM < 1/2$. Then, we can resort to the following inequality [Flum and Grohe, 2006, Lemma 16.19]

$$\sum_{l=0}^{L} \binom{k}{l} \le 2^{H_2(L/k)k}, \quad \text{if } L/k \le 1/2, \tag{116}$$

where $H_2(p) := -p\log(p) - (1-p)\log(1-p)$. Therefore, for all $k \ge \max\{C_M(\kappa/\lambda), n\}$, we obtain

$$(1-\underline{p})^{\lfloor\frac{k}{n}\rfloor - \ell_k\left(\frac{\lambda}{\kappa}\right)} \sum_{l=0}^{\ell_k\left(\frac{\lambda}{\kappa}\right)} \binom{\lfloor\frac{k}{n}\rfloor}{l} \le (1-\underline{p})^{\lfloor\frac{k}{n}\rfloor - \ell_k\left(\frac{\lambda}{\kappa}\right)} 2^{H_2\left(\frac{\lfloor\ell_k(\lambda/\kappa)\rfloor}{\lfloor k/n\rfloor}\right)\lfloor k/n\rfloor} \tag{117}$$

$$\le (1-\underline{p})^{\lfloor\frac{k}{n}\rfloor - 2nM\lfloor\frac{k}{n}\rfloor} 2^{H_2(2nM)\lfloor\frac{k}{n}\rfloor} \tag{118}$$

$$= \left[(1-\underline{p})^{1-2nM} 2^{H_2(2nM)}\right]^{\lfloor\frac{k}{n}\rfloor}, \tag{119}$$

since $(1-\underline{p}) \in [0,1)$ and $H_2(p)$ is an increasing function for $p \in (0, 1/2)$.

We define $\eta_M := (1-\underline{p})^{1-2nM} 2^{H_2(2nM)}$. Note that for $M = 0$, we have $\eta_0 = (1-\underline{p})2^{H_2(0)} = (1-\underline{p}) \in [0,1)$. By the continuity of $\eta_M$ in $M$, there exists $M \in (0, 1/4n)$ such that $\eta_M \in (0,1)$. By (111), (112), (115), and (119), we obtain

$$\mathbb{P}\left\{\#_k s' \le \ell_k\left(\frac{\lambda}{\kappa}\right) \mid k^\dagger - k = \kappa\right\} \le \eta_M^{\lfloor\frac{k}{n}\rfloor}, \quad \forall k \ge \max\left\{C_M\left(\frac{\kappa}{\lambda}\right), n\right\}. \tag{120}$$

Its sum over $k \ge 0$ (corresponding to the inner sum in (109)) is bounded from above by

$$\sum_{k=0}^{\infty} \mathbb{P}\left\{\#_k s' \le \ell_k\left(\frac{\lambda}{\kappa}\right) \mid k^\dagger - k = \kappa\right\} \overset{(a)}{\le} C_M\left(\frac{\kappa}{\lambda}\right) + n + \sum_{k=0}^{\infty} \eta_M^{\lfloor\frac{k}{n}\rfloor} \tag{121}$$

$$\overset{(b)}{\le} C_M\left(\frac{\kappa}{\lambda}\right) + n + \sum_{k=0}^{\infty} \eta_M^{\frac{k}{n}-1} \tag{122}$$

$$\overset{(c)}{=} C_M\left(\frac{\kappa}{\lambda}\right) + n + \frac{1}{\eta_M(1-\eta_M^{1/n})}, \tag{123}$$

where $(a)$ follows since any probability is bounded from above by one and $\max\{C_M(\kappa/\lambda), n\} \le C_M(\kappa/\lambda) + n$; $(b)$ follows since $\lfloor k/n \rfloor > k/n - 1$ and $\eta_M \in (0,1)$; and $(c)$ follows since $\eta_M^{1/n} \in (0,1)$ when $n > 0$.

Based on (123), the sum (109) is bounded from above by

$$\sum_{\kappa=1}^{\infty} \mathbb{P}\left\{k^{\dagger} - k = \kappa\right\} \sum_{k=0}^{\infty} \mathbb{P}\left\{\#_k s' \le \ell_k\left(\frac{\lambda}{\kappa}\right) \mid k^{\dagger} - k = \kappa\right\}$$

$$\le \sum_{\kappa=1}^{\infty} \mathbb{P}\left\{k^{\dagger} - k = \kappa\right\}\left(C_M\left(\frac{\kappa}{\lambda}\right) + n + \frac{1}{\eta_M(1-\eta_M^{1/n})}\right) \tag{124}$$

$$\le \sum_{t=1}^{\infty} \mathbb{P}\left\{(t-1)n < k^{\dagger} - k \le tn\right\}\left(C_M\left(\frac{tn}{\lambda}\right) + n + \frac{1}{\eta_M(1-\eta_M^{1/n})}\right), \tag{125}$$

where the last inequality follows since $C_M(\cdot)$ is non-decreasing. Either Assumption 2 or 2' yield that

$$\mathbb{P}\left\{(t-1)n < k^{\dagger} - k \le tn\right\} \le (1-\underline{p})^{t-1}. \tag{126}$$

Therefore, we can bound (125) from above by

$$\sum_{t=1}^{\infty} (1-\underline{p})^{t-1}\left(C_M\left(\frac{tn}{\lambda}\right) + n + \frac{1}{\eta_M(1-\eta_M^{1/n})}\right). \tag{127}$$

Since $C_M(\cdot)$ is a polynomial function by Assumption 1-ii, the ratio test, i.e.,

$$\lim_{t\to\infty} \frac{(1-\underline{p})^t\left(C_M\left(\frac{(t+1)n}{\lambda}\right) + n + \frac{1}{\eta_M(1-\eta_M^{1/n})}\right)}{(1-\underline{p})^{t-1}\left(C_M\left(\frac{tn}{\lambda}\right) + n + \frac{1}{\eta_M(1-\eta_M^{1/n})}\right)} = 1 - \underline{p} < 1, \tag{128}$$

yields that (126) is convergent for any $\lambda > 0$. Therefore, we obtain (104), which completes the proof.
$\square$

**Lemma 3**. *Consider any trajecttory of* (63) *and let* $x(t) := (q^1(t), q^2(t), \pi^1(t), \pi^2(t))$. *Then the candidate function* $V(\cdot)$, *as described in* (64), *satisfies*

- $V(x(t')) < V(x(t))$ *for all* $t' > t$ *if* $V(x(t)) > 0$,
- $V(x(t')) = 0$ *for all* $t' > t$ *if* $V(x(t)) = 0$.

*Proof:* Fix an arbitrary solution $(q^1(t), q^2(t), \pi^1(t), \pi^2(t))$ to the o.d.e. (63) for some arbitrary initial point, and define the functions $L : [0, \infty) \to \mathbb{R}$ and $H : [0, \infty) \to \mathbb{R}$ by

$$L(t) := \sum_{i=1,2} \max_{\mu \in \Delta(A^i)} \left\{\mu \cdot q^i(t) + \tau\nu^i(\mu)\right\} - \lambda\zeta \tag{129}$$

$$H(t) := \|q^1(t) - Q^1\pi^2(t)\|^2 + \|q^2(t) - Q^2\pi^1(t)\|^2. \tag{130}$$

Then, we obtain

$$V(q^1(t), q^2(t), \pi^1(t), \pi^2(t)) = [L(t)]_+ + H(t). \tag{131}$$

Note that $H(t)$ is a continuous, differentiable and non-negative function. Its time derivative is given by

$$\frac{dH(t)}{dt} = -2H(t). \tag{132}$$

If $\zeta = 0$, then $L(\cdot)$ reduces to the Lyapunov function introduced by Harris [1998] for continous-time best response dynamics in zero-sum strategic-form games, and therefore, $V(\cdot)$ is a Lyapunov function function for (63).

Suppose that $\zeta > 0$. Note that $L(\cdot)$ is also a continuous and differentiable function, but it can also be negative (when $\zeta > 0$) and the last term of (129), i.e., $-\lambda\zeta$, is time-invariant. For notational simplicity, let $\overline{\pi}^i(t) := \overline{\mathrm{Br}}^i(q^i(t), \tau)$. Based on the envelope theorem (which can be invoked due to

the smoothness of the optimization argument induced by $\nu^i(\cdot)$), the time derivative of $L(t)$ is given by

$$\frac{dL(t)}{dt} = \sum_{i=1,2} \overline{\pi}^i(t) \cdot \frac{dq^i(t)}{dt} \tag{133}$$

$$= \sum_{i=1,2} \overline{\pi}^i(t) \cdot \left( Q^i \overline{\pi}^{-i}(t) - q^i(t) \right) \tag{134}$$

$$= -\sum_{i=1,2} \left( \overline{\pi}^i(t) \cdot q^i(t) + \tau \nu^i(\overline{\pi}^i(t)) \right)$$

$$+ \overline{\pi}^1(t)^T \left( Q^1 + (Q^2)^T \right) \overline{\pi}^2(t) + \tau \sum_{i=1,2} \nu^i(\overline{\pi}^i(t)). \tag{135}$$

By definition, $\overline{\pi}^i(t) \in \Delta(A^i)$ and $\nu^i(\overline{\pi}^i(t)) \leq \log(|A^i|)$. Therefore, we have

$$\overline{\pi}^1(t)^T \left( Q^1 + (Q^2)^T \right) \overline{\pi}^2(t) + \tau \sum_{i=1,2} \nu^i(\overline{\pi}^i(t)) \leq \zeta, \tag{136}$$

where $\zeta$ is as described in (65). Correspondingly, the time derivative of $L(t)$ is bounded from above by

$$\frac{dL(t)}{dt} < -\sum_{i=1,2} \left( \overline{\pi}^i(t) \cdot q^i(t) + \tau \nu^i(\overline{\pi}^i(t)) \right) + \lambda \zeta \tag{137}$$

$$= -L(t), \tag{138}$$

where the strict inequality follows since $\lambda > 1$ and $\zeta > 0$. This yields that $L(t)$ is *strictly* decreasing whenever $L(t) \geq 0$. Therefore, $\{q^1(t), q^2(t), \pi^1(t), \pi^2(t) : L(t) \leq 0\}$ is a positively invariant set for any trajectory. In other words, if $L(t') \leq 0$ for some $t'$, then $L(t'') \leq 0$ for all $t'' > t'$. Therefore, by (131), the time-derivatives (132) and (138) yield that $V(\cdot)$ is a Lyapunov function, which completes the proof. $\qquad\square$

## F   Proof of Corollary 1 to Theorem 1

**Corollary 1.** *Suppose that player $-i$ follows an (asymptotically) stationary strategy $\{\tilde{\pi}_s^{-i} \in \text{int}\Delta(A_s^i)\}_{s \in S}$ while player $i$ adopts the learning dynamics described in Table 1, and Assumption 1 holds. Then, the asymptotic behavior of the value function estimate $\{\hat{v}_{s,k}^i\}_{k \geq 0}$ is given by*

$$\limsup_{k \to \infty} \left| \hat{v}_{s,k}^i - \max_{\pi^i} v_{\pi^i, \tilde{\pi}^{-i}}^i(s) \right| \leq \epsilon \xi^i g(\gamma), \quad \text{under Assumption 2,} \tag{139a}$$

$$\lim_{k \to \infty} \left| \hat{v}_{s,k}^i - \max_{\pi^i} v_{\pi^i, \tilde{\pi}^{-i}}^i(s) \right| = 0, \qquad \text{under Assumption 2'} \tag{139b}$$

*for all $s \in S$, w.p. 1, where $\xi^i := \max_{s' \in S} \left\{ \log(|A_{s'}^i|) \right\}$ and $g(\cdot)$ is as described in Theorem 1.*

*Furthermore, the asymptotic behavior of the weighted averages $\{\hat{\pi}_k^i\}_{k \geq 0}$, described in Theorem 1, is given by*

$$\limsup_{k \to \infty} \left( \max_{\pi^i} v_{\pi^i, \tilde{\pi}^{-i}}^i(s) - v_{\hat{\pi}_k^i, \tilde{\pi}^{-i}}^i(s) \right) \leq \epsilon \xi^i h(\gamma), \quad \text{under Assumption 2,} \tag{140a}$$

$$\lim_{k \to \infty} \left( \max_{\pi^i} v_{\pi^i, \tilde{\pi}^{-i}}^i(s) - v_{\hat{\pi}_k^i, \tilde{\pi}^{-i}}^i(s) \right) = 0, \qquad \text{under Assumption 2',} \tag{140b}$$

*for all $s \in S$, w.p. 1, where $h(\gamma)$ is as described in Theorem 1, i.e., these weighted-average strategies converge to near or exact best-response strategy, depending on whether Assumption 2 or 2' hold.*

*Proof:* The proof follows from the observation that Theorem 1 can be generalized to the scenarios where nature draws $r_s^i(a^1, a^2) \in [-D, D]$ and $p(s'|s, a^1, a^2)$ depending on a random event in a rather straightforward way since it only introduces a stochastic approximation error that is a square integrable Martingale difference sequence. For example, player $i$ receives $r_s^i(\omega_k, a^1, a^2)$ with random

event $\omega_k \in \Omega$ and $r_s^i(\omega_k, a^1, a^2) \in [-D, D]$ for all $\omega \in \Omega$ and state transitions are governed by the kernel $p(s'|s, \omega_k, a^1, a^2)$ while $\omega_k \to \omega_o$ as $k \to \infty$ with probability 1.

Since player $-i$ follows an asymptotically stationary strategy $\{\tilde{\pi}_s^{-i}\}_{s \in S}$ almost surely, we can view player $-i$ as nature and its action at stage $k$, $a_k^{-i}$, as the random event $\omega_k$. Then, it reduces into a single-player game. We can still invoke Theorem 1 if we introduce an auxiliary player $i'$ that has a single action at every state without loss of generality. This completes the proof. $\qquad\square$

## G  Additional Simulation Setup

We consider a larger-scale case with $|S| = 20$ states and $|A_s^i| = 10$ actions per state. The discount factor $\gamma = 0.5$. The reward functions are chosen randomly in a way that $r_s^1(a^1, a^2) \propto \bar{r}_{s,a^1,a^2}$ for $s \in S$, where $\bar{r}_{s,a^1,a^2}$ is uniformly drawn from $[-1, 1]$. Then, $r_s^1(a^1, a^2)$ is normalized by $\max_{s,a^1,a^2}\{r_s^1(a^1, a^2)\}/2$ so that $|r_s^i(a^1, a^2)| \leq R = 2$ for all $(i, s, a^1, a^2)$. For the state transition dynamics $p$, we construct two cases, **Case 3** and **Case 4** by randomly generating transition probabilities, in a way that they satisfy Assumptions 2-i and 2'-i, respectively. For **Case 3** and **Case 4**, we choose the temperature parameter as $\tau_c = \max\{\epsilon, \tau_c'\}$ and as $\tau_c'$ in (12), respectively, with $\epsilon = 2 \times 10^{-2}$, $\bar{\tau} = 0.1$. For both cases, we choose the stepsizes $\alpha_c = 1/c^{0.9}$ and $\beta_c = 1/c$ with $\rho_\alpha = 0.9$, $\rho_\beta = 1$, and $\rho = 0.85$ for the $\tau_c'$ in (12). The simulation results are illustrated in Figure 3. Note that as the number of states is large, the plot becomes very dense and cluttered if both players' curves are plotted, together with the standard-deviation bar-area, as in Figure 3. We thus only plot an example trial, with only Player 2's curves, and the summation of the value function estimates. The convergence of Player 1's value function estimates can be deduced accordingly. It is seen from Figure 3 that our theory can be corroborated by simulations even for this larger-scale case.