# OpenReview forum: "Decentralized Q-learning in Zero-sum Markov Games"
_NeurIPS.cc/2021/Conference — NeurIPS 2021 Poster_

### Official Review · Reviewer_nE2G · 2021-07-16

**Rating:** 5
**Confidence:** 2

**Summary:**

The paper studies the convergence of decentralized q-learning in two-player zero-sum games. The theoretical conclusion of the paper is sound. My main concern is regarding the originality and significance of this paper in comparison to a recent ICML 2021 paper.

**Ethics Review Area:**

["I don’t know"]

**Limitations And Societal Impact:**

Not apply

**Main Review:**

* The paper studies the convergence of decentralized q-learning in two-player zero-sum games. There seems a recent ICML 2021 paper studies the same learning in games setting and reaches almost the same result: http://proceedings.mlr.press/v139/guo21a/guo21a.pdf. So one concern is about the originality and significance of this paper.

* In addition, one of the key assumptions in deriving the convergence results is that every state is visited infinitely many times --> so that the global value function estimation \hat{v} can be an unbiased estimator. But the appearance of the state seen by agent 1 is affected by the action/stratedgies of agent 2. What if agent 2 is a dummy player that takes the same action or an adversarial player? Actually, the state is not visited infinitely many times (and many states may not be visited) is a key challenge in actually training MARL. The other paper ICML 2021 didn't make this assumption - and this may be one of the reasons that they only have a stochastic convergence guarantee instead of the prob 1 results achieved in this paper.

* The author may want to explain the experiment setup in further details. In Figure 2(a, b) There are 5 lines for each agent in each sub-plot, are they corresponding to 5 different zero-sum games? what are the game setup and parameters?

* Figure 1 is confusing - it might be helpful to add more explanations or adjust the presentation to be more clear, and link the definition of global and local Q-functions to the corresponding equations.

**Time Spent Reviewing:**

1.5

---

> ### Author Response · Authors · 2021-08-10
> **Response to Reviewer nE2G**
>
> We thank the reviewer for the comments. As stated in the General Response, the ICML paper had not appeared when we submitted our paper. After carefully reviewing the ICML paper, we believe that there are significant differences between the two papers (see response to Reviewer 1VzG). We also answer your questions as follows:
>
>
> 1. We respectfully disagree that "almost the same result" is obtained here, and "the originality and significance" is a concern. The two papers are substantially different in terms of motivation (even though both papers do not require the observation of the opponent's action, we focus more on being "rational" and "convergent" simultaneously, with “symmetric” and “natural” update-schemes), algorithm types, analysis (and proof techniques), convergence results, assumptions, etc. We are motivated by presenting a “natural” learning dynamics of the “best-response”-type, inspired by the literature of “learning in games” theory. These kinds of learning dynamics have been studied extensively in learning in “static matrix/normal-form games”, while their definitions and convergence properties have been an open problem that our paper managed to address for the first time. See response to Reviewer 1VzG for a more detailed comparison to the results in [1].
>
>
> 2. The analysis tools used in our paper and the ICML paper are fundamentally different: we are based on stochastic approximation theory while they used mirror descent-type of optimization theory. Correspondingly, the assumptions in both settings are usually very different. In the analysis of asynchronous stochastic approximation, the infinitely-often visitation of the states is a very standard and mild assumption, and has also been used even in proving the “single-agent” Q-learning’s convergence. This essentially ensures that a good estimate of the Q-value can be obtained for all the states.
>
> On the other hand, even in the game setting, our assumption is standard, and finds sources (and is even weaker than some assumptions) in the literature (see lines 259-267). In general, if the opponent is dummy or even adversary, the convergence property of our learning dynamics is not clear. However, as we responded above (especially see responses to Reviewer 1VzG), the beauty of being "rational" is that the decentralized and self-interested agent has the incentive to follow this update rule, as if she is solving a single-agent problem, without even knowing the existence of the opponent. On a different but related note, in the ICML paper the reviewer mentioned, some state-action visitation assumption is also needed, see their Assumption 4.3 and the discussions after it.
>
>
> 3. The five lines correspond to five "states" in the stochastic game. We have the details in the appendix. Due to space limitation, we deferred the experimental details to the appendix. We will add them back to the main text in the final version. We will also improve the presentation of Figure 1, and link the definitions to the equations.
>
>
> Again, we appreciate the helpful comments from the reviewer, and hope that we have fully addressed the concerns.

---

### Official Review · Reviewer_j6QZ · 2021-07-16

**Rating:** 8
**Confidence:** 3

**Summary:**

This paper introduces a two-timescale variant of decentralised Q-learning for two-player zero-sum Markov games (without terminal states). Convergence of the algorithm is shown under several assumptions, and example experiments are provided to illustrate this convergence in practice.

**Limitations And Societal Impact:**

These aspects are addressed in Section 5.

**Main Review:**

I thank the authors for their detailed response to my queries, and maintain my rating. I'm glad the authors will add more detail around the experiment to the main paper. Interesting points were raised in other reviews regarding e.g. degree of decentralisation that algorithm achieves given the need for similar time-scale updates between the players - further discussion of these points in a later version of  the paper would be useful.

---

This is a strong paper with an important central contribution. The paper is also generally very clearly written, and in my opinion the authors have achieved an excellent balance in the main paper between describing the algorithm, required assumptions, proof sketch, and illustrative experimental results. In addition, the work is well contrasted against a variety of different approaches to MARL in zero-sum Markov games.

I have not had time to check all proofs in the appendix line-by-line, but have checked Propositions 1-3 and found these to be correct.

Some minor comments are given below:

Statement of Proposition 3: I don't think "Assumption 2" or "Assumption 2'" are defined as terms in the paper - does Assumption 2 mean 2-i and 2-ii together?

Section 4: I appreciated the inclusion of a proof-of-concept implementation and experiment here. However, at present there is not really enough detail provided in the main paper to take much from the diagrams, and the reader is essentially required to consult the appendix to understand the significance of what is illustrated. If space allows, I think this section could be improved by including sketch details of the environment (and algorithm settings), perhaps at the cost of not illustrating convergence for both assumption sets in the main paper. As a very minor point, the last x-axis tick label in Fig 2a seems to have been cut off.

Questions for the authors:
 * Regarding the clipped learning in Line 2 of the Table 1, presumably any choice of upper-bound (in the range (0,1]) to clip at would yield a convergent algorithm (i.e. with Prop. 2 still holding for this modified algorithm, and the remainder of the argument then going through). Is there any particular reason to select 1 as the upper-bound here, other than the largest value that still guarantees boundedness of the iterates?
 * Do the authors have a sense of how important is the choice of entropy regulariser in Table 1? I.e. could a convergence proof hold for other classes of regulariser with the properties of strong concavity, unbounded gradients at simplex boundary etc.

Minor notational comments:
There are a few minor notational issues in the main paper:
 * I found #s quite odd notation, in contrast to something like N(s). Potentially the notation used should also be something like N_k(s_k) rather than N(s_k) (or #_k s_k rather than # s_k in current notation), to make clear the dependence on k of the visit count. Without this, statements such as Proposition 3 look odd, since in "# s -> infty as k -> infty", the dependence of # s on k is not clear. This notation is used to some extent (e.g. proof of Prop 3), but would be clearer if used throughout.
 * The notation is a little imprecise in places: Eqn (1): expectation written over actions, but not states?
 * There are also some typos to be corrected in the main paper and appendix e.g. "ith" -> "with" Line 607.

**Time Spent Reviewing:**

4

---

> ### Author Response · Authors · 2021-08-10
> **Response to Reviewer j6QZ**
>
> We really appreciate the very positive comments from the reviewer.
>
> Response to the minor comments:
> - Yes, Assumption 2 refers to Assumption 2-i and 2-ii together.
> - Thanks for the suggestion. We will move the details of the simulation setup back to the main text in the final version (when we have an extra page). We will also edit the x-axis.
>
>
>
> Response to the questions for the authors:
> - There is no particular reason to choose the threshold as 1 than it appears natural to us, since it ensures that the iterates always get updated to a convex combination of the last iterate and the new observation. Analysis also holds for another upper bound that can ensure boundedness.
> - Thanks for raising a good point. We choose entropy as a smooth function since it is common in the literature and leads to an analytical-form expression also known as “soft-max” exploration. We use this analytical form expression in Assumptions 1 and 2 to characterize explicit conditions on the step sizes. Other regularizers, being strongly concave and unbounded gradients at simplex boundary, could also be used, but the conditions on step sizes guaranteeing convergence to an equilibrium would have been different.
>
> Finally, thank you very much for the careful reading and for pointing out the minor notational issues. We will address them to improve the readability in the final version.
>
> Thank you again for the thorough review of our paper. We hope all your minor concerns have been satisfactorily addressed.

---

### Official Review · Reviewer_p9zY · 2021-07-17

**Rating:** 7
**Confidence:** 2

**Summary:**

This paper presents a fully decentralized Q-learning algorithm for Markov games.  The individual ("local") Q-function is tracked separately from a state-value function, with updates to the Q-function taken according to a TD-error computed using the state-value function (of the next state).  Crucially, these two function are learned at differing timescales, making the action-reward value for each state "appear" stationary.  The algorithm is demonstrated to converge to Nash equilibrium (or the neighborhood of a Nash equilibrium, depending on the strength of assumptions made) in zero-sum Markov games when both players use it, both theoretically and numerically.

**Limitations And Societal Impact:**

Limitations are directly and adequately addressed, including societal impact.

**Main Review:**

This paper gives a very elegant solution to a persistently vexing problem in MARL.  The solution has the potential to extend beyond the simple setting initially studied by the paper, and could have a substantial impact.  The paper is very well structured, and gives a clear motivation and intuition for each aspect of its results.

#### Additional comments ####

* It seems like Theorem 1 is a consequence of Corollary 1 more than the reverse.  In particular, Corollary 1's statement does not appear to depend upon the zero-sum property of the game at all, whereas Theorem 1 is a straightforward consequence of Corollary 1 given that the game has the zero-sum property.  This could point toward extensions beyond the zero-sum case.  E.g., stochastic games in which each stage game is a potential game seems like a natural next step, since better-response dynamics are guaranteed to converge to an equilibrium, and corollary 1 seems to point to a strong connection with better-response dynamics.

* All of the statements of Theorem 1 and related results refer to global learning rates $\{\alpha_c\}$ and $\{\beta_c\}$ rather than agent-specific sequences.  Is this without loss of generality, or does the result of Theorem 1 rely upon both agents using common learning rates?  If yes, then this is still more uncoupled than a case where agents must agree on who is the "fast" learner and who is the "slow" one, but it still requires some coordination to agree on the learning schedule.

* On a related note, is this algorithm more vulnerable than standard independent Q-learning approaches to exploitation by a sophisticated opponent who knows the learning algorithm being used?  I.e., if I know the opponent is using a particular learning schedule, can I exploit their value-estimates' being out of date by a particular non-stationary strategy?

* I found the convergence described in Figure 2 to be somewhat surprising; it appears that one or the other agent always converges to a stage-game-suboptimal strategy in each stage game, rather than a "natural" equilibrium in which each player plays some equilibrium of the stage game at each stage.  (This latter equilibrium is guaranteed to exist, even when the players have different discount factors; it would give every state a discounted value of 0).  Is this a consequence of the requirement to play each action with a softmax probability (i.e., when there is a tie in expected continuation value between two actions, they must be played with equal probability if they also have the same expected reward)?  Or is there something deeper going on here?  Or is this just happenstance of which equilibrium was selected?



**Time Spent Reviewing:**

3.5

---

> ### Author Response · Authors · 2021-08-10
> **Response to Reviewer p9zY**
>
> We thank the reviewer for appreciating our work "gives a very elegant solution to a persistently vexing problem in MARL", "could have a substantial impact", and "very well structured". We respond to your questions as follows:
>
> 1. Although Corollary 1 can be seen as a more general result, it makes a strong assumption that the opponent has an asymptotically stable strategy. On the other hand, Theorem 1 shows this in the zero-sum case. A counterpart of Theorem 1 for games beyond zero-sum such as identical interest games would be of great interest, and is indeed one of our ongoing research directions. We appreciate the reviewer pointing it out.
>
>
> 2. We really appreciate the reviewer raising this subtle and interesting point. Our results are devised for a “symmetric” scenario where players follow the same step sizes. However, our analysis could be straight-forwardly generalized to the cases where the agents’ step sizes follow the same assumptions as we made in the paper, but with possibly different “values”, respectively, as long as the “order” of the stepsizes are common (i.e., $\alpha_c^1/\alpha_c^2 \to 1$ and $\beta_c^1/\beta_c^2 \to 1$ as $c\to \infty$). In other words, our results hold if there is no “timescale” separation between “agents”. Using a timescale-separated stepsizes between agents (the only case that our results do not cover) makes it less decentralized, since this “asymmetric” learning dynamics requires some coordination between agents to determine who is faster and who is slower. On the other hand, [Daskalakis et al. 2020] have shown that independent “policy gradient” methods could converge to an equilibrium in zero-sum stochastic games if they use this “two-timescale learning rate” where one agent updates slower. Whether this could also be the case in our “Q-learning” framework or not is an interesting future research question.
>
>
> 3. We are not exactly sure if we fully understand the question. But in general, we agree that if the opponent is allowed to use any non-stationary strategy, and knows the learning rule of the agent (though by “rationality”, a self-interested opponent might not have the incentive to do so), it is possible to break the convergence of the agent. However, we think this is a general issue that the “independent Q-learning” (and more generally “decentralized” learning dynamics) would also be vulnerable to, rather than that our learning dynamics is “more” vulnerable.
>
>
> 4. We are not exactly certain about the points being asked. What Figure 2 shows is the convergence performance to the Nash equilibrium value at different states. At each state, it IS a Nash equilibrium of the stage game. We are not very sure what did the reviewer mean by “one or the other agent always converges to a stage-game-suboptimal strategy in each stage game”, and “even when the players have different discount factors; it would give every state a discounted value of 0”, since the discounted factor $\gamma$ is common for both players. We are happy to discuss more about this question, if the reviewer could kindly provide more details.
>
>
> Again, we would like to thank the reviewer for the careful reading, and for appreciating the contribution of our work.

---

### Official Review · Reviewer_KoGL · 2021-07-17

**Rating:** 5
**Confidence:** 3

**Summary:**

The authors propose an independent Q-learning algorithm for two-player zero-sum Markov games. Asymptotic convergence results are obtained for the cases where both (or only one) players both follow the proposed algorithm.

**Limitations And Societal Impact:**

Yes

**Main Review:**

Writing quality and clarity are okay. It's nice to have figures to illustrate the Q-functions. I do have some questions and concerns regarding the theoretical results.

1. Asymptotic convergence: Having asymptotic convergence is not a strong enough result. Given the works [1, 2], can the authors get a rate of convergence? If not, what are the algorithmic/technical barriers that make getting convergence rates fundamentally harder?

2. Rationality property: I think the authors are trying to argue  in ​"rationality property" (Corollary 1) that a player would learn good response even if the opponent does not follow the proposed algorithm. But my concern here is that,

(1) When considering asymptotic convergence, letting the opponent using a stationary strategy almost surely actually means you are fixing the opponent's strategy throughout the learning process. How does this differ from learning optimal policy in the single-agent setting

(2) Do we not allow the opponent to choose whatever strategy he wants in each iteration? If that's the case, aren't we still asking the two players to collaborate in a sense that the opponent should either follow the algorithm or keep using the same strategy "almost surely"?

(3) Is best-in-hindsight response to arbitrary strategy sequence possible?

In my opinion, the above concerns make the convergence results in this paper less appealing that the typical convergence results with rates, and weaken the claim that the algorithm is fully "decentralized".

[1] https://arxiv.org/pdf/2011.01868.pdf

[2] https://arxiv.org/pdf/2010.15020.pdf

**Time Spent Reviewing:**

3

---

> ### Author Response · Authors · 2021-08-10
> **Response to Reviewer KoGL**
>
> First, we thank the reviewer for the positive comments.
>
> 1. Our focus in this paper was an asymptotic result, since establishing convergence for the learning dynamics considered here do not follow from standard two-timescale stochastic approximation theory, and requires significant efforts. And yes, we are fully aware of the literature [1,2] (and in fact cited [2], and will also cite [1]). However, it is challenging to build upon either of these results since:
>
> (1) [1] follows some standard assumption in two-timescale stochastic approximation in the literature. However, the assumptions in [1] regarding the uniform Lipschitz continuity of such best-response $H$, the uniform strong monotonicity of the operators, the synchronous update, and the independent assumption on the noises, generally do not hold in our case. In fact, this was exactly the reason we could not directly apply Borkar's results on two-timescale stochastic approximation. Particularly, in our learning dynamics, the update at faster timescale may not even be stable for a given iterate of the slow update since the associated stage game may deviate from the zero-sum structure and there are many counterexamples showing that best-response-type of dynamics including fictitious play may not converge to an equilibrium in a general-sum game, e.g., see [Shapley, 1964]. Here, the stage games can deviate from the zero-sum structure since the players’ continuation payoff estimates do not necessarily sum to zero when they update them “independently” and “simultaneously” through their smoothed best responses (while keeping the update “independent”, “simultaneous”, and “symmetric” were the key desiderate to make our learning dynamics “natural” and “rational” for self-interested agents). To address this, we first approximate our discrete-time update at the fast timescale through its limiting ordinary differential equation only to characterize its limit set in terms of the sum of continuation payoff estimates via a new “Lyapunov function”. Based on this characterization, we analyze the convergence properties of the sum of continuation payoff estimates at the discrete time by exploiting some contraction properties and show that they go to zero asymptotically, i.e., stage games become zero-sum asymptotically. Finally, we combine the characterization of the limit set and the result that sum of continuation payoff estimates goes to zero to show convergence of slow iterates again at the discrete time. While conducting this analysis, the “asynchronous” update of the iterates is an additional challenge to invoke the classical stochastic approximation methods and necessitate alternative approaches such as [Tsitsiklis, 1994]. In sum, highly non-trivial efforts have been devoted to prove even asymptotic convergence in our setting (which is necessary to ensure our desirable “decentralized” learning dynamics works). As explained before, our paper provides the first convergence guarantee for such natural learning dynamics, and we leave the finite-sample analysis under our “non-standard” assumptions of the dynamics as a future direction (see line 330).
>
>
> (2) [2] considers a finite-horizon (episodic) setting (one can reset after each episode, while ours is an online and restless setting), and cares about some notion of "weak regret", which does not necessarily lead to Nash equilibrium convergence in the zero-sum setting (as [2] considers a more ambitious goal of handling even "unknown" type of games, from only a single-agent’s perspective). We have carefully discussed the difference in lines 109-113, with more details in the appendix. Note that the regret-analysis techniques used in [2] are fundamentally different from the stochastic approximation theory used in our paper. It would be interesting to generalize the ideas in [1] to the infinite-horizon discounted setting, which is also one of our future directions.
>
>
> 2. Rationality:
> (1) Yes, in the asymptotic setting, when simply “letting” the opponent use a stationary strategy almost surely would lead to a single-agent learning setting, and our “rationality” property ensures that the agent would use the best-response strategy to the opponent using a stationary strategy. However, note that, we do not “impose” the opponent to use such a stationary strategy, nor require any coordination among agents to do so. Rather, our results prove that each agent, by playing a myopic best-response-type update (which is natural and sensible for each agent to follow), converges to a stationary strategy, and jointly converges to a Nash equilibrium. This is exactly the beauty of being rational: the agent has the incentive to follow the update rule, since it would reduce to the single-agent setting when the opponent is "stationary". See more justifications for this desideratum in Bowling and Veloso [2001], Busoniu et al. [2008], and Wei et al. [2021]. Note that, this property does not hold in many existing MARL algorithms.
>
> (2) and (3): for "arbitrary" strategy sequences of the opponent, the convergence is not clear. Thus, it “seems” that we only allow the case where the opponent either follows the algorithm update, or uses asymptotically stationary strategies. However, following the response to point (1), the “rational” property of our learning dynamics ensures that the opponent, as a self-interested agent, would have the motivation to follow the dynamics. In other words, the equilibrium is a natural "outcome" of some learning dynamics that the agents tend to follow (without any coordination to do so). See more discussions on “rationality” in the response to Reviewer 1VzG.
>
>
> Overall, we appreciate the valuable comments from the reviewer. We hope that we have satisfactorily addressed your concerns.

---

### Official Review · Reviewer_1VzG · 2021-07-26

**Rating:** 5
**Confidence:** 3

**Summary:**

The paper studies the Q-learning algorithm in the multi-agent setting, under the assumption that each agent observes its action and reward only while being completely unaware of the other agent's actions and rewards (decentralized setting). The asymptotic convergence of the proposed algorithm is proved under two different sets of assumptions regarding the algorithm hyperparameters. An illustrative experiment is presented.

**Limitations And Societal Impact:**

This is a mainly theoretical paper. I do not foresee any societal impact.

**Main Review:**

***Major***
- My main concern is about the novelty of the work compared with the recent work [1]. Both the present paper and [1] consider the Q-learning algorithm in the multi-agent decentralized setting, with no communication between the agents. Besides the fact that [1] considers a single-timescale rule and the present paper a two-timescale rule, it seems to me that the results of [1] are more general/stronger. In particular:
1) The present paper limits to the finite state case, whereas [1] uses function approximation and, therefore, can be applied to continuous state spaces (although the theoretical results are provided under the assumption that the function approximator allows representing the Q-function)
2) The present paper provides an asymptotic convergence guarantee, while [1] manages to derive a finite-sample guarantee.

Compared to [1], the present paper proves that the proposed algorithm is "rational", which seems a not too demanding requirement. Indeed, if the opponent is playing a stationary strategy, the problem reduces to single-agent RL.

- Assumption 2-ii. The value of the temperature converges to \epsilon in this assumption. This means that the agent will never play a deterministic policy. Instead, a small exploration will always be preserved. Does this imply that the algorithm will converge to an \epsilon-Nash equilibrium rather than a Nash equilibrium?

- Experiment. The experiment seems a little sacrificed there. The explanation is very synthetic, and nothing about the employed environment is reported in the main paper. I suggest considering moving the experiment to the supplementary material.


***Minor***
- Line 185: the subscript k is introduced here, but its meaning is explained only later
- Table 1: shouldn't this be considered an "Algorithm" rather than a "Table"? Missing full stop at the end of the caption.
- Is there any relation between the requirements of Assumption 1 and the classical Robbins-Monro conditions?

***Overall***
The paper is written in good English, and the presentation is overall clear. My main concern, as reported above, is about the novelty of the work compared with [1]. I would greatly appreciate it if the authors could elaborate on the substantial differences between their work and [1]. For this reason, at present, I opt for a borderline score.

[1] Guo, Hongyi, Zuyue Fu, Zhuoran Yang, and Zhaoran Wang. "Decentralized Single-Timescale Actor-Critic on Zero-Sum Two-Player Stochastic Games." In International Conference on Machine Learning, pp. 3899-3909. PMLR, 2021.

**Time Spent Reviewing:**

2.5

---

> ### Author Response · Authors · 2021-08-10
> **Response to Reviewer 1VzG**
>
> We thank the reviewer for the helpful comments. First of all, we emphasize that at the time of submitting our paper, reference [1] had not yet appeared anywhere online, and we were not aware of the results. Further, after reviewing the paper, we believe there are significant differences between the two, as detailed below:
>
> 1. The algorithm in [1] is not Q-learning (as the reviewer mentioned), but actor-critic. They are fundamentally quite different. Our goal is to present a natural, rational, and decentralized learning dynamics that can be implemented by self-interested agents, based on the best-response type of dynamics in the “learning in (matrix) games” literature, which is intuitive for the agent to follow.
>
>
> 2. [1] considers the function approximation setting, while we consider the basic and fundamental tabular setting. However, it is not clear if the results in [1] would strictly “cover” the tabular setting we considered, given the different assumptions and update-rules.
>
>
> 3. [1] has non-asymptotic convergence, however, under stronger assumptions and different update-rules; to name a few: 1) The i.i.d. sampling assumption from the stationary state-action distribution; 2) Projection onto ball with bounded radius R, to ensure stability of the iterates and quantify the norm of the iterates; 3) A "double-loop" update rule, i.e., line 3 in Algorithm 1 requires a subroutine "BestResponse" where player 2 updates multiple steps, with the iterates of player 1 "fixed", which makes it less "decentralized" and "rational" than our "simultaneous" update-rules, and is not symmetric. Their update scheme thus requires coordination between the players, and requires them to be fully aware of the fact that they are playing a game; while our learning dynamics is symmetric, coordination-free, and the player is agnostic of the game; 4) Finite Concentration (or more commonly "Concentrability") Coefficient in Assumption 4.1 for "an arbitrary sequence of policies", which seems a strong assumption even in the “tabular” case; 5) Their update rule is based on actor-critic algorithms (policy-based RL approach), while our learning dynamics are Q-learning based (value-based RL approach). More importantly, for researchers who know both types of analysis, the proof techniques (stochastic approximation) and the challenges confronted in our paper are fundamentally different from those in [1], requiring different assumptions and different novelties. Hence, it is not clear if one is definitely more “general/stronger” than the other.
>
> Being "rational": In contrast to viewing it as “a not too demanding requirement”, we believe that “being rational” is an important advantage of our algorithm, which is a desired property of MARL algorithms and in the Economics literature of "learning in games", and has not been paid attention to in many existing MARL works. It is aimed to address the following fundamental question in “learning in games”: whether a game theoretical equilibrium could be realized as a result of non-equilibrium adaptation of self-interested agents that are learning simultaneously. One of the most basic and intuitive adaptation rules for self-interested agents is to “best-respond” to their “estimate” of the opponent’s strategy. The “rational” property considered here ensures such best-response adaptation is achieved when players asymptotically converge to a stationary strategy, and is a necessary condition for convergence to an equilibrium. Nevertheless, many existing MARL algorithms do not satisfy this property (for “all” the players, not just one of them, e.g., as in [1] with a “double-loop” update scheme).
>
> The point of being "rational" is that, even if the agent is myopic and is unaware of the opponent (which is exactly the case in the "decentralized" setting), she still has the motivation to follow the update-rule. To illustrate the importance of this property, consider the rock-paper-scissors game, where an equilibrium is the case where both players play all actions uniformly. On the other hand, if the opponent is playing only rock, then playing all actions uniformly is not the best response. Therefore, when players play the same game repeatedly, players will look for “predicting” the opponent’s play to make the best response to it. When both players look for such “non-equilibrium adaptation” (which is of their own interests), this “may or may not” result in an equilibrium. However, an answer affirmative (or not) is of importance to understand the underlying theory for the interactions of such self-interested agents. Therefore, this question has been studied extensively in the theory of “learning in games” literature, by showing the convergence of best-response (or fictitious play)-type of learning dynamics, in normal-form (matrix) games. In fact, it has been a long-standing problem whether such “decentralized” adaptation “leads to an equilibrium” in stochastic/Markov games considered here, since the introduction of fictitious play by Brown [1949] and stochastic games by Shapley [1953]. This is mainly due to the “non-stationarity” issue from a single-agent’s perspective, when all players are learning simultaneously and independently, in a decentralized fashion.
>
> Our paper addresses exactly this question, and provides for the first time a convergence guarantee for such learning dynamics. Our learning dynamics do not involve imposing certain rules on agent behavior to “make” the environment stationary (while [1] does so by imposing the “double-loop” update scheme). Instead, it provides a natural decentralized learning dynamics motivated by the “behavioral considerations” in which self-interested agents form beliefs of the opponent strategy (indirectly through local Q-functions) and the continuation payoff, and play a (smoothed) best-response. In this regard, our dynamics differ fundamentally from the “algorithmic” approaches studied in the literature, with the goal of just “computing the Nash equilibrium”, possibly with “coordination” among agents (including [1]).
>
> Assumption 2-II: Yes, under this assumption, as proved in our Theorem 1, (14a), the convergence to Nash equilibrium value is not exact (up to some epsilon-order error).
>
> Experiments: Thanks for the suggestion. We were short of space, and will move the details back to the main text in the final version.
>
> Finally, regarding the minor issues: $k$ has been defined as time index in Sec. 2.1, and we will define it explicitly in the final version; we will change “table” and use full stop; Yes, Assumption 1-i corresponds to the non-summable condition in the Robbins-Monro conditions. Note that the condition on the sum of step sizes’ squares is included in Assumption 2-ii while Assumption 2’-ii poses a different condition. Also, Assumption 1-ii poses additional conditions on step sizes to ensure two-timescale learning dynamics.
>
> Again, we thank the reviewer for the useful comments, and hope that our responses have fully addressed your concerns.

---

### Author Response · Authors · 2021-08-10
**General Response**

We first would like to thank all the reviewers and the area chairs for the very helpful reviews. One common concern was regarding the possible overlap with Guo et al., ICML 2021. However, at the time when we submitted our paper, this ICML paper was not publicly available (note that the NeurIPS submission deadline is before the ICML camera-ready submission deadline, and is also before the time when the ICML proceedings are available). In the final version of our article we will of course comment thoroughly on the relationship between the two contributions. In fact, having now carefully read the ICML paper, it is clear that our proposed learning dynamics and results (in terms of algorithm design, analysis techniques, settings, motivations) are significantly different from those in that paper (to be detailed below). We would greatly appreciate the reviewers and the area chair to re-evaluate our paper after reading other reviews and our responses.

---

> ### Comment · Area_Chair_uSjN · 2021-08-26
> **Comparison with Guo et al. (ICML 2021)**
>
> Dear authors of Paper 6399, I have no doubt that your paper and the paper by Guo et al. (ICML 2021) are independent, concurrent works. The discussion on your paper is still on, but we all agree that, in the case of acceptance, you need to add a subsection clearly stating the differences between your work and that by Guo et al. (ICML 2021). I ask you to produce such a subsection and post here on OpenReview by three days (let me know if 3 days are not sufficient). This section would not guarantee that the paper will be accepted as there are also other concerns about which we are discussing, but it can help. Furthermore, we would have a clear idea on how you would change the paper.
>
> Thanks.

---

> > ### Author Response · Authors · 2021-08-29
> > **Comparison with Guo et al. (ICML 2021)**
> >
> > We would like to first thank the area chair and the reviewers for appreciating the independency and concurrency of the two works, as well as their significant differences. We would be happy to summarize the differences between the two papers as follows, which will also be included in the final version of our paper:
> >
> >
> > After submitting the first draft of our paper, we became aware of an independent and concurrent work of Guo et al., 2021, which also studied a decentralized learning setting in zero-sum Markov games. Though sharing the key word “decentralized” in the title, there are significant differences between the two works, as summarized below:
> >
> >
> > 1. **Motivation**: In Guo et al., 2021, being “decentralized” is defined as “each player not knowing the opponent’s action”, to “protect the privacy”, and the goal is to “compute” the Nash equilibrium of the game; in contrast, in our work, in addition to “being oblivious to the opponent’s action”, we also allow no “coordination” among agents, so that each agent can simply run the learning dynamics “individually”, without even being aware of the existence of the opponent. The agents in our setting are considered as self-interested decision-makers, who seek to adapt to the opponent's play by inferring it from the rewards received without seeing the opponent's actions. The Nash equilibrium, on the other hand, is the result that “emerge” naturally when both agents follow this self-interested learning dynamics (and we have proved this). Finally, as our learning dynamics are oblivious to the opponent and are adaptive to the opponent, we expect it to converge beyond the zero-sum setting (e.g., the identical-interest setting), which is one of our ongoing research directions. In contrast, the algorithm in Guo et al. is specifically developed for the zero-sum setting. These motivations differ fundamentally from Guo et al., 2021 (and thus creates very different technical challenges, as detailed below).
> >
> >
> > 2. **Learning dynamics (Algorithms)**: The algorithm in Guo et al., 2021, is actor-critic, which is a type of policy-based RL method; the learning dynamics in our work is Q-learning based, which belongs to value-based RL methods. More importantly, the update-rule in Guo et al., 2021, is of “double-loop” form, in the sense that it fixes the iterate of Player 1 while updating Player 2’s policy, so that a “best-response” policy of Player 2 can be obtained. This is an asymmetric update-rule, and requires coordination between agents. In contrast, our learning dynamics are “symmetric”, without such a double-loop coordination, where each agent simply runs her own Q-learning dynamics.
> >
> >
> > 3. **Assumptions and Results**: Guo et al., 2021, considers a function approximation setting, and assumes that: 1) the “double-loop” update can be implemented by the agents in the decentralized setting; 2) the concentration (or more commonly “Concentrability”) coefficient is finite (Assumption 4.1), for “an arbitrary sequence of policies”; 3) samples are drawn i.i.d. from the stationary state-action distribution; 4) projection of the iterates onto some ball with radius $R$, to ensure the iterates’ stability; and 5) zero approximation error of the Bellman operator (Assumption 4.2). Under these assumptions, non-asymptotic convergence results were established. In contrast, our work considers a fundamental tabular setting, and without making these assumptions (1-4), with however asymptotic convergence guarantees. Note that, the individual and simultaneous update rules of ours have created highly non-trivial challenges in our analysis. It is also not clear if some of the assumptions above, e.g., finite concentrability assumption, trivially hold in our tabular setting. Finally, we note that both papers require some type of reachability assumption (their Assumption 4.3 and our Assumption 2-I and 2’-I). With these significantly different assumptions, it is not clear if one paper’s result is stronger than (or subsumes) the other’s.
> >
> >
> > 4. **Analysis techniques (Technical novelty)**: The analyses, as well as the technical novelties in both papers are not comparable. The analysis technique in Guo et al., 2021, is a mirror-descent type of analysis, based on the convergence analysis of policy gradient (and actor-critic) algorithms in single-agent RL. It more of an optimization-type analysis. The techniques in our paper, however, are based on stochastic approximation theory, a classic technique in showing the convergence of Q-learning. The challenges we need to address (our technical novelties) mainly lie in constructing a Lyapunov function and stability of the iterates, within this non-standard two-timescale stochastic approximation setting, with asynchronous updates. Such challenges would not be encountered in the analysis of Guo et al., 2021, making the technical novelties of the two papers fundamentally different.
> >
> >
> > Thank you very much again,
> >
> > The Authors

---

### Decision · Program_Chairs · 2021-09-27

**Decision:**

Accept (Poster)

**Comment:**

Almost all the Reviewers agree that this paper is worth being published. The potential overlap with the ICML 21 paper raised initially serious concerns by the Reviewers. However, all the Reviewers agree that this paper and the ICML 21 paper are concurrent. I also believe that these two papers present several differences and the authors clearly discuss these differences in the rebuttals. I invite the authors to add a section in their paper in which the differences with ICML 21 are discussed. I also invite the authors to take into seriously consideration the following concerns raised by Reviewer KoGL and to clarify these issues in the paper:

*The "fully" decentralized claim is still questionable to me after reading the response. To some extent, it does make sense for agents to follow "myopic" best-response dynamics. But still, assuming every one of them being at the same level of "myopia" is sort of collaborating. And the Corollary 1 does not really tell any meaningful thing in terms of the algorithm's ability to exploit opponents that do not follows such dynamics.*

Frankly, I don't think that *assuming every one of them being at the same level of "myopia" is sort of collaborating*, but it could be useful that the authors clarify better this point in their paper.